# Ataxin-7 and Non-stop coordinate SCAR protein levels, subcellular localization, and actin cytoskeleton organization

Veronica Cloud[1], Ada Thapa[1], Pedro Morales-Sosa[1], Tayla M Miller[1], Sara A Miller[1], Daniel Holsapple[1], Paige M Gerhart[1], Elaheh Momtahan[1], Jarrid L Jack[1], Edgardo Leiva[1], Sarah R Rapp[1], Lauren G Shelton[2], Richard A Pierce[2], Skylar Martin-Brown[2], Laurence Florens[2], Michael P Washburn[2,3], Ryan D Mohan[1]*

[1]University of Missouri - Kansas City, Kansas City, United States; [2]Stowers Institute for Medical Research, Kansas City, United States; [3]Department of Pathology and Laboratory Medicine, University of Kansas Medical Center, Kansas City, United States

*For correspondence:
MohanRD@umkc.edu

Competing interests: The authors declare that no competing interests exist.

**Abstract** Atxn7, a subunit of SAGA chromatin remodeling complex, is subject to polyglutamine expansion at the amino terminus, causing spinocerebellar ataxia type 7 (SCA7), a progressive retinal and neurodegenerative disease. Within SAGA, the Atxn7 amino terminus anchors Non-stop, a deubiquitinase, to the complex. To understand the scope of Atxn7-dependent regulation of Non-stop, substrates of the deubiquitinase were sought. This revealed Non-stop, dissociated from Atxn7, interacts with Arp2/3 and WAVE regulatory complexes (WRC), which control actin cytoskeleton assembly. There, Non-stop countered polyubiquitination and proteasomal degradation of WRC subunit SCAR. Dependent on conserved WRC interacting receptor sequences (WIRS), Non-stop augmentation increased protein levels, and directed subcellular localization, of SCAR, decreasing cell area and number of protrusions. *In vivo*, heterozygous mutation of SCAR did not significantly rescue knockdown of Atxn7, but heterozygous mutation of Atxn7 rescued haploinsufficiency of SCAR.
DOI: https://doi.org/10.7554/eLife.49677.001

## Introduction

The S̲pt A̲da G̲cn5 A̲cetyltransferase (SAGA) transcriptional coactivator complex is an approximately 1.8 MDa, modular, multi-protein complex, bearing two enzymatic subunits (Gcn5 acetyltransferase and Non-stop deubiquitinase) housed in separate modules of the complex (*Lee et al., 2011*). Within SAGA, the Ataxin-7 (Atxn7) amino-terminus anchors the deubiquitinase module (DUBm) to the larger complex (*Lee et al., 2009*). Mutation of *Atxn7* and *non-stop* leads to aberrant gene expression, defective non-homologous end joining, decreased immune response, reduced replicative lifespan, neurodegeneration, blindness, tumorigenesis, and cancer (*Glinsky et al., 2005*; *Weake et al., 2008*; *McCormick et al., 2014*; *Lang et al., 2011*; *Bonnet et al., 2014*; *Furrer et al., 2011*; *Li et al., 2018*; *Mohan et al., 2014a*).

The progressive retinal and neurodegenerative disease S̲pinoc̲erebellar A̲taxia type 7̲ (SCA7) is caused by CAG trinucleotide repeat expansion of the *ATXN7* gene, resulting in polyglutamine (polyQ) expansion at the amino terminus of the Atxn7 protein (*Giunti et al., 1999*; *David et al., 1997*). SCA7 disease is characterized by progressive cone-rod dystrophy leading to blindness, and progressive degeneration of the spine and cerebellum (*Martin, 2012*; *Garden, 1993*).

In *Drosophila*, loss of *Atxn7* leads to a phenotype similar to overexpression of the polyQ-expanded amino terminal truncation of human Atxn7 (*Latouche et al., 2007*) – reduced life span, reduced mobility, and retinal degeneration (*Mohan et al., 2014a*). Without Atxn7, the *Drosophila* DUBm is released, enzymatically active, from SAGA. Once released, the module acts as a gain-of-function, leading to reduced ubiquitination of H2B. Similarly, the mammalian DUBm binds and deubiquitinates substrates without Atxn7 *in vitro*, albeit at a lower rate (*Lan et al., 2015*; *Yang et al., 2015*). Consistent with Non-stop release and over activity mediating the *Drosophila* Atxn7 loss-of-function phenotype, reducing *non-stop* copy number alleviates lethality associated with loss of Atxn7 (*Mohan et al., 2014a*).

Non-stop is a critical mediator of retinal axon guidance and important for glial cell survival (*Weake et al., 2008*; *Martin et al., 1995*). Interestingly, abnormal ubiquitin signaling in the nervous system contributes to a number of genetic and spontaneous neurological and retinal diseases, including SCA3, SCAR16, Alzheimer's, Parkinson's, ALS, and Huntington's disease (*Petrucelli and Dawson, 2004*; *Campello et al., 2013*; *Mohan et al., 2014b*). Few functions are known for the DUBm beyond deubiquitination of H2Bub, H2Aub, shelterin, and FBP1 (*Atanassov and Dent, 2011*; *Atanassov et al., 2009*). In polyQ-Atxn7 overexpression models, sequestration of the DUBm may result in reduced deubiquitinase activity on critical substrates. Interestingly, both increased and decreased H2Bub have been shown to hinder gene expression, and imbalances (whether up or down) in ubiquitination are observed in retinal and neurological diseases (*Ristic et al., 2014*). Together, these data suggest more needs to be understood about Atxn7-mediated regulation of DUBm function.

To better understand the Non-stop-Atxn7 regulatory axis, we set out to identify substrates for Non-stop and determine the consequence of Atxn7 loss on Non-stop/substrate interactions. To this end, we purified Non-stop-containing complexes, fractionated them by size, and tested enzymatic activity to identify novel complexes bearing enzymatically active Non-stop. This revealed the functionally active DUBm associates with protein complexes distinct from SAGA. Mass spectrometry revealed Arp2/3 complex and Wiskott-Aldrich syndrome protein (WASP)-family verprolin homologous protein (WAVE) Regulatory Complex (WRC) members suppressor of extracellular cAMP receptor (cAR) (SCAR), HEM protein (Hem), and specifically Rac1-associated protein 1 (Sra-1), as interaction partners of the independent DUBm. These complexes function together to initiate actin branching (*Kurisu and Takenawa, 2009*). Pull-down and immunofluorescence verified interaction and revealed extensive colocalization between Non-stop and WRC subunit SCAR.

In flies, loss of Atxn7 led to increased interaction between Non-stop and SCAR. In cells and in flies, loss of Atxn7 resulted in a 2.5-fold increase in SCAR protein levels, while loss of Non-stop led to 70% decrease in SCAR protein levels. Spatiotemporal regulation of SCAR is essential for maintaining cytoskeletal organization. A constant ubiquitination-proteasomal degradation mechanism contributes to establishing and maintaining SCAR protein amount. Mutating the Non-stop enzymatic pocket to increase affinity for ubiquitin led to increased interaction with SCAR. An alternative enzymatic pocket mutation decreasing affinity for ubiquitin reduced interaction with SCAR. When Non-stop was knocked-down in cells, the amount of ubiquitinated SCAR increased. Loss of SCAR in Non-stop mutants was rescued by proteasome inhibition, suggesting Non-stop counters SCAR polyubiquitination and proteasomal degradation.

An emerging class of proteins, important for neural function and stability bear W̲RC i̲nteracting r̲eceptor s̲equences (WIRS) which contribute to direct interaction and spatiotemporal regulation of SCAR (*Chen et al., 2014*). Close examination of Non-stop amino acid sequence revealed conserved WIRS motifs in *Drosophila* and human Non-stop orthologues. Mutating these decreased Non-stop-SCAR interaction, but allowed Non-stop to assemble into SAGA.

Overexpression of Non-stop increased total SCAR protein levels approximately 5-fold. Interestingly, SCAR protein levels increased in cellular compartments where Non-stop increased, including in the nucleus. When SCAR protein levels were increased and relocalized by augmenting Non-stop, local F-actin amounts increased too, leading to cell rounding and fewer cell protrusions. Mutation of Non-stop WIRS motifs reduced Non-stop ability to increase SCAR protein, F-actin, and to reduce cellular protrusions. Although SCAR heterozygosity was not sufficient to rescue defective retinal axon targeting upon Atxn7 knockdown, loss of Atxn7 was sufficient to rescue loss of F-actin in SCAR heterozygotes.

## Results

### Non-stop interacts with multi-protein complexes distal from SAGA

Loss of Atxn7 releases Non-stop from SAGA resulting in a deubiquitinase gain-of-function, reducing ubiquitination of histone H2B (*Mohan et al., 2014a*). The resultant retinal and neurodegenerative phenotype is rescued by reducing *non-stop* yet it is inconsistent with phenotypes arising from mutations in other regulators of H2B ubiquitination, such as *Drosophila* homolog of RAD6 (Dhr6) (the E2 ubiquitin conjugating enzyme), Bre1 (E3 ubiquitin ligase), or Ubiquitin-specific protease 7 (USP7) (H2B deconjugating enzyme) (*van der Knaap et al., 2005*; *van der Knaap et al., 2010*; *Tsou et al., 2012*; *Haddad et al., 2013*; *Mohan et al., 2010*; *Koken et al., 1991*; *Bray et al., 2005*). Therefore, we sought novel interaction partners for Non-stop (*Figure 1A*). To this end, an unbiased biochemical screen was deployed to specifically identify protein complexes stably interacting with enzymatically active DUBm (*Figure 1B*). *Drosophila* S2 cells were stably transfected with a pRmHA3-based inducible expression vector to produce epitope-tagged Non-stop-2xFLAG-2xHA (Non-stop-FH) (*Mohan et al., 2014a*; *Cherbas et al., 2011*; *Guelman et al., 2006*; *Kusch et al., 2004*; *Suganuma et al., 2008*; *Bunch et al., 1988*). Whole cell extracts were prepared from these cell lines and immunoblotted to verify they expressed amounts of Non-stop comparable to wild-type cells (*Figure 1C*, *Figure 1—figure supplement 1*).

Non-stop-containing protein complexes were purified from nuclear extracts via tandem affinity purification of their epitope tags: FLAG, then HA (*Suganuma et al., 2010*). The complexity of these samples were assessed by silver stain (*Figure 1D*). Isolated complexes from SAGA subunits Atxn7 and Will Decrease Acetylation (WDA) are shown for comparison (*Mohan et al., 2014a*). Although similar in profile to Atxn7 and WDA purifications, Non-stop-containing complexes were unique, indicating Non-stop has novel interaction partners.

To separate SAGA from other Non-stop-interacting protein complexes, purified complexes were separated by size using Superose 6 gel filtration chromatography (*Figure 1E*, upper) (*Kusch et al., 2003*). For comparison purposes, data for Atxn7 and Ada2B are reproduced here (*Mohan et al., 2014a*). Atxn7 and Ada2b are SAGA-specific subunits. Their elution from the gel-filtration column indicates those fractions contain SAGA (*Mohan et al., 2014a*; *Kusch et al., 2003*). Non-stop eluted in additional fractions (*Figure 1E*, upper).

To determine which fractions contained enzymatically active Non-stop, the ubiquitin-AMC deubiquitinase activity assay was performed, in which the fluorescent probe AMC is initially quenched by conjugation to ubiquitin but fluoresces when released by deubiquitination, permitting an indirect measurement of the relative amount of deubiquitination occurring in each sample (*Figure 1E*, lower) (*Köhler et al., 2010*; *Samara et al., 2010*). In complexes purified through Atxn7, peak deubiquitinase activity was detected around 1.8MDa, consistent with SAGA complex and previous purification, fractionation, and mass spectrometry of Atxn7, which is reproduced here for purposes of comparison (*Mohan et al., 2014a*; *Kusch et al., 2003*; *Lee et al., 2011*). In Non-stop purifications, however, deubiquitinase activity resolved into three major peaks. The major activity peak resolved at about 1.8 MDa, along with SAGA, and was labeled 'Group 1'. A secondary peak, 'Group 2', was resolved centering approximately 669 kDa, which was interpreted to be non-SAGA large multi-protein complexes. A final peak with the lowest enzymatic activity, centering about 75 kDa, was labeled 'Group 3.' The DUBm, consisting of Non-stop, Sgf11, and E(y)2 are predicted to have a molecular mass of 88 kDa, meaning this last peak was the DUBm interacting with very small proteins, if any. Silver staining of a single fraction at the center of Group 2 revealed the fractionation procedure resolved complexes into a pattern distinct from the total input (*Figure 1E*, right).

The three fractions comprising the center of each peak were combined, and MudPIT mass spectrometry was performed to identify their protein constituents (*Washburn et al., 2001*). MudPit proteomics provides a distributed normalized spectral abundance factor (dNSAF) which acts as a reporter for relative protein amount within a sample (*Washburn et al., 2001*). Consistent with previous characterizations of SAGA, Group 1 contained SAGA complex subunits from each of the major (HAT/TAF/SPT/DUB) modules (*Mohan et al., 2014a*; *Kusch et al., 2003*; ; *Lee et al., 2011*; *Setiaputra et al., 2015*). Group 2 was selected for further interrogation because it was predicted to contain non-SAGA large multi-protein complexes. Group 2 contained the DUBm: Non-stop, Sgf11, and E(y)2 – indicating the DUBm remained intact during purification and fractionation (*Köhler et al., 2010*; *Samara et al., 2010*; *Lee et al., 2011*). Histone H2A and H2B were also detected in Group 2,

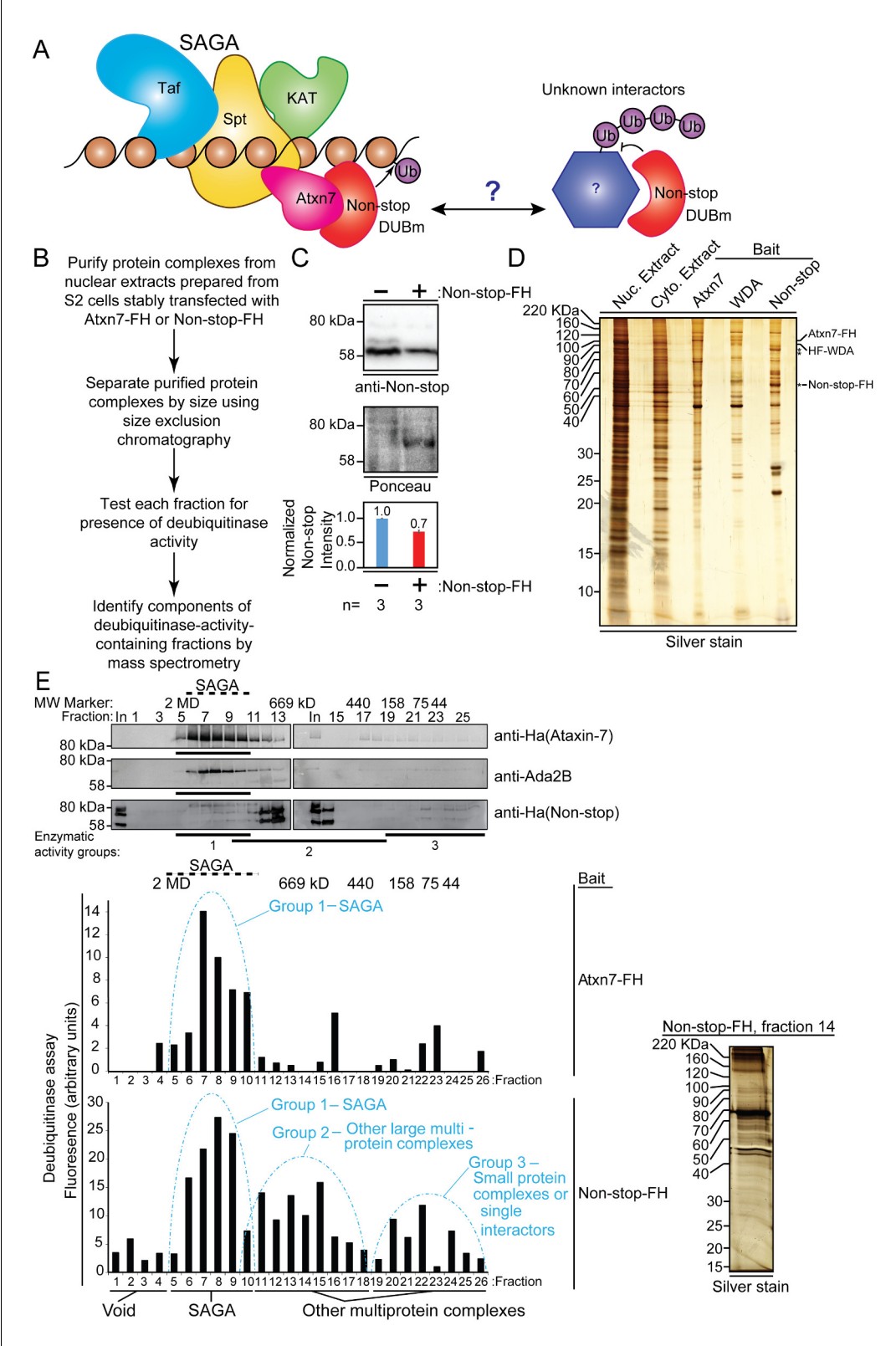

**Figure 1.** Non-stop (Not) co-purifies with Spt Ada Gcn5 acetyltransferase (SAGA) complex and additional multi-protein complexes. (**A**) Working model: The Non-stop-containing SAGA deubiquitinase module (DUBm) functions distally from SAGA. Non-stop is anchored to SAGA by Ataxin-7 (Atxn7) but is released to interact with unknown partners. (**B**) Scheme to identify Non-stop interactors. (**C**) Characterization of Non-stop-2xFLAG-2xHA (FH)-expressing S2 cell lines. Stably transfected cells were treated with 10 uM copper sulphate show comparable levels of Non-stop expression in parental and stably

*Figure 1 continued on next page*

eLIFE Research article

Cell Biology | Neuroscience

Figure 1 continued

transfected cell lines. Parental S2 cells containing no plasmid compared to S2 cells stably transfected with Non-stop plasmid. Quantitation of anti-Non-stop immunoblot signal intensities normalized to Ponceau total protein loading control. Error bars represent standard error. (D) Analysis of purified complexes. Non-stop-containing complexes were analyzed by silver staining. Atxn7 and Will Decrease Acetylation (WDA)-containing complexes are shown for comparison. (E) Protein complexes, purified and fractionated as described in B were analyzed by immunoblotting to observe relative elution by size (top, Atxn7 and Ada2B recreated from *Mohan et al., 2014b*) followed by measure of deubiquitinase activity contained in each fraction as assayed by ubiquitin-AMC assay in which increased fluorescence correlates directly with deubiquitination activity. The complexity of eluted fractions was analyzed and Group 2 peak fraction, number 14, is shown (right).

DOI: https://doi.org/10.7554/eLife.49677.002

The following figure supplement is available for figure 1:

**Figure supplement 1.** Anti-Non-stop antibody specifically recognizes Non-stop.

DOI: https://doi.org/10.7554/eLife.49677.003

suggesting the DUBm co-purified with known substrates, further confirming conditions were suitable for preservation and identification of endogenous interactions (*Figure 2A*) (*Zhao et al., 2008*; *Henry et al., 2003*). Furthermore, Non-stop dNSAF was not over-represented relative to Sgf11 and E(y)2, further confirming Non-stop bait protein was not expressed beyond physiological levels (*Figure 2A*) (*Zhang et al., 2010*).

## Non-stop co-purifies with Arp2/3 and WRC complexes

Notably, within Group 2, Arp2/3 complex (all subunits) and WRC subunits SCAR, Hem, and Sra-1 were represented in high ratio to DUBm members, as determined by dNSAF (*Figure 2A*) (*Pollard and Beltzner, 2002*; *Chen et al., 2010*).

WRC complex activates Arp2/3 to promote actin branching, which is important for a range of cellular activities including cell migration, cell adhesion, exocytosis, endocytosis, maintenance of nuclear shape, and phagocytosis (*Zallen et al., 2002*; *Takenawa and Miki, 2001*; *Alekhina et al., 2017*; *Navarro-Lérida et al., 2015*; *Vishavkarma et al., 2014*; *Verboon et al., 2015*). Non-stop and Atxn7-containing complexes were purified from nuclear extracts, where their function in gene regulation is best characterized. WRC and Arp2/3 function in the nucleus is more enigmatic. Although Arp2/3 and WRC have demonstrated nuclear functions, their roles in actin branching in the cytoplasm are better understood (*Navarro-Lérida et al., 2015*; *Verboon et al., 2015*; *Yoo et al., 2007*; *Rawe et al., 2004*; *Miyamoto et al., 2013*). SAGA, WRC, and Arp2/3 complexes are all critical for proper central nervous system function, making the regulatory intersection for these complexes intriguing for further study (*Zallen et al., 2002*; *Meyer and Feldman, 2002*; *Chou and Wang, 2016*; *Dumpich et al., 2015*; *Kessels et al., 2011*; *Irie and Yamaguchi, 2004*).

In *Drosophila*, SCAR is particularly important for cytoplasmic organization in the blastoderm, for egg chamber structure during oogenesis, axon development in the central nervous system, and adult eye morphology (*Zallen et al., 2002*). To further characterize the relationship between Non-stop, Atxn7, and WRC, we turned to the BG3 cell line (ML-DmBG3-c2, RRID:CVCL_Z728). These cells were derived from the *Drosophila* 3rd instar larval central nervous system. Biochemical and RNAseq profiling of these cells indicate they express genes associated with the central nervous system. Their morphology is similar to that of *Drosophila* CNS primary cultured cells (*Cherbas et al., 2011*; *Ui et al., 1994*).

To verify interaction between Non-stop and WRC, BG3 cells were transiently transfected with either Non-stop-FH or SCAR-FLAG-HA (SCAR-FH) and immunoprecipitated using anti-FLAG resin. Anti-HA antibody was used to verify pull down of the tagged protein. A previously verified anti-SCAR antibody confirmed the presence of SCAR in Non-stop-FH immunoprecipitate (*Figure 2B*) (*Verboon et al., 2015*; *Rodriguez-Mesa et al., 2012*; *Cetera et al., 2014*; *Tran et al., 2015*; *Verboon and Parkhurst, 2015*). Reciprocal immunoprecipitation of SCAR-FH utilizing an anti-FLAG resin followed by immunoblotting for Non-stop using an anti-Non-stop antibody confirmed SCAR-FH associated with endogenous Non-stop (*Figure 2C*) (*Mohan et al., 2014a*).

To visualize the scope of Non-stop and WRC colocalization, indirect immunofluorescence was used to localize endogenous SCAR and Non-stop in BG3 cells. BG3 cells were plated on concanavalin A coated coverslips giving them an appearance similar to S2 cells plated on concanavalin A (*Rogers et al., 2003*) (*Figure 2D*). SCAR localized in a characteristic pattern with high concentrations

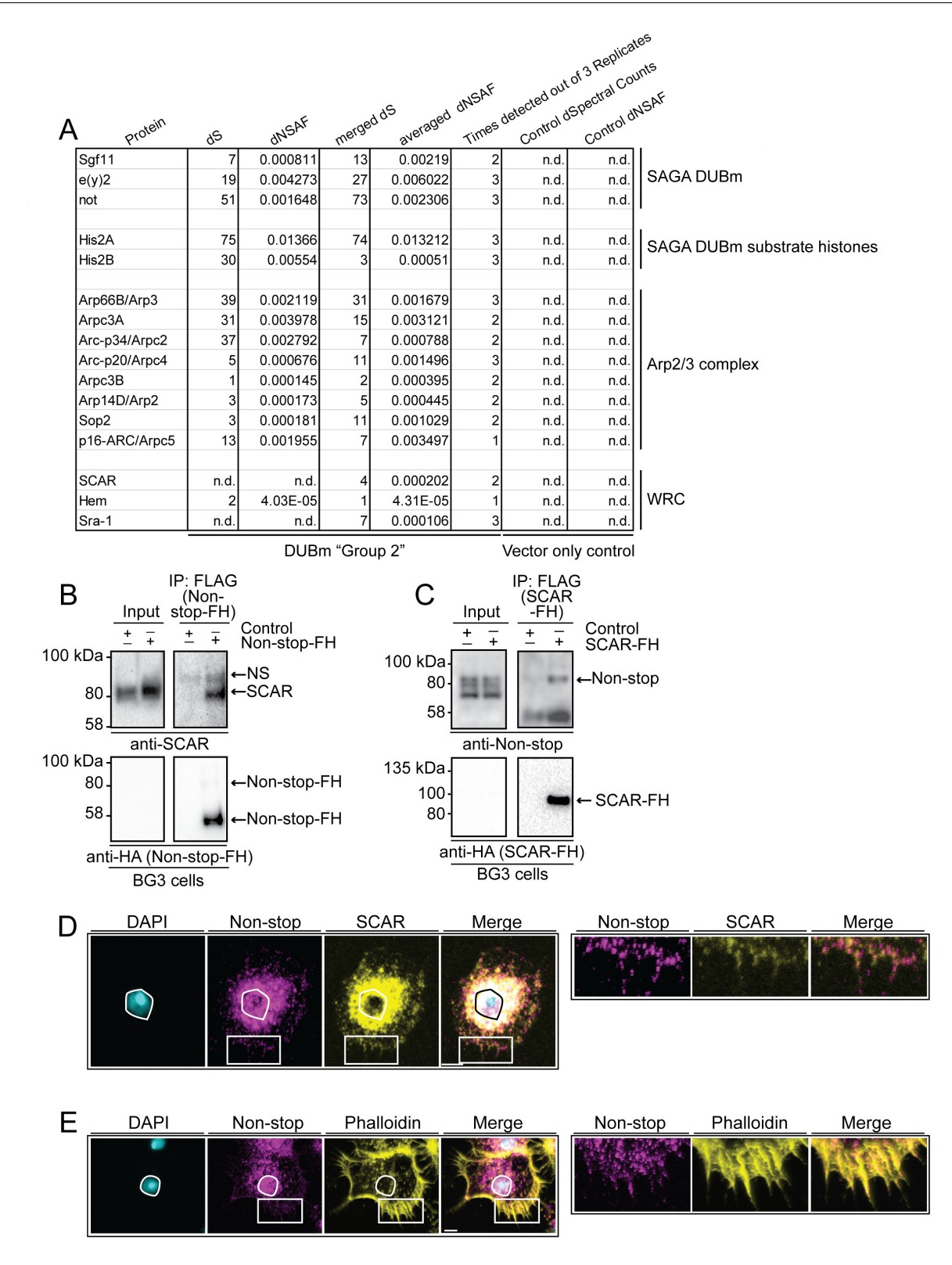

**Figure 2.** WAVE regulatory complex (WRC) and Arp2/3 interact with Non-stop. (**A**) Mass spectrometry analysis of Group 2 fractions revealed Arp2/3 and WRC complexes stably interact with Non-stop. None of these proteins were identified using vector only Control purifications. (**B and C**) Reciprocal pull-down verifies interaction between Non-stop and WRC subunit suppressor of extracellular cAMP receptor (cAR) (SCAR). (**B**) BG3 cells were transfected with an expression vector for Non-stop-2xFLAG-2xHA (Non-stop-FH) or no vector (Control). Recombinant Non-stop was

*Figure 2 continued on next page*

*Figure 2 continued*

immunoprecipitated using anti-FLAG affinity resin and purified interactors analyzed by immunoblotting for the presence of endogenous SCAR. Anti-HA immunoblots verified the presence of Non-stop-FH bait. NS marks a non-specific band. (C) Pull-down was performed as in B, but with SCAR-FLAG-HA (SCAR-FH) expression vector. Immunoprecipitated proteins were probed by immunoblotting to detect endogenous Non-stop. Anti-HA immunoblots verified the presence of SCAR-FH bait. (D) Endogenous Non-stop and SCAR occupy similar subcellular regions. Immunofluorescence of SCAR and Non-stop in BG3 cells. Cells were immunostained with anti-Non-stop (magenta), anti-SCAR (yellow), and DAPI (white). Scale bar is 10 µM. Inset is enlarged to the right. (E) Endogenous Non-stop and F-actin are found in similar subcellular regions. Representative images of BG3 cells immunostained with anti-Non-stop (magenta), phalloidin (F-actin)(yellow), and DAPI (DNA)(white). Scale bar is 10 µM. Inset is enlarged to the right. Images in D and E were adjusted with contrast limited adaptive histogram equalization using the ImageJ CLAHE algorithm (*Zuiderveld, 1994*).

DOI: https://doi.org/10.7554/eLife.49677.004

around the nucleus and at the cell periphery (*Rogers et al., 2003*). Non-stop localized similarly to SCAR, but also further within nuclei. Examination of the spatial relationship between Non-stop and F-actin, using phalloidin, revealed Non-stop overlapping with portions of F-actin/phalloidin (*Figure 2E*) (*Cooper, 1987*).

## Atxn7 and Non-stop coordinate SCAR protein levels

Atxn7 loss releases Non-stop from SAGA and is lethal, although, lethality is rescued by reducing Non-stop gene copy number. This implies a class of Non-stop target(s), which associate more with Non-stop in the absence of Atxn7. Furthermore, it raises the possibility that a subset of these may contribute to toxicity resulting from Atxn7 loss (*Mohan et al., 2014a*). To determine if interaction between Non-stop and SCAR is regulated by Atxn7, endogenous Non-stop was immunoprecipitated from either wild-type or *Atxn7*-mutant 3$^{rd}$ instar larval whole cell extracts. Immunoprecipitates were immunoblotted to detect endogenous SCAR protein. In *Atxn7* mutant extracts, approximately 1.4 times more SCAR was present per Non-stop captured (*Figure 3A*, right). This suggests Atxn7 retains a pool of Non-stop that can be freed to interact with SCAR.

If Non-stop was acting as a SCAR deubiquitinase to counter proteolytic degradation, then loss of Atxn7 would lead to increased SCAR protein. Conversely, reducing Non-stop would lead to less SCAR protein. To test these hypotheses, RNAi was used to knock down Non-stop or Atxn7 transcripts in cell culture and SCAR protein levels were observed by immunoblotting. BG3 cells were soaked in dsRNA targeting LacZ (negative control), Non-stop, or Atxn7. After six days, denaturing whole cell protein extracts were prepared from these cells and immunoblotted to detect SCAR. Ponceau S total protein stain was used as loading control to verify analysis of equal amounts of protein (*Romero-Calvo et al., 2010*; *Lee et al., 2016*). Protein extracts from Non-stop knockdown cells consistently showed approximately 70% decrease in SCAR protein levels relative to LacZ control knockdown. Consistent with a model in which Atxn7 restrains Non-stop deubiquitinase activity, cells treated with Atxn7-targeting RNAs showed 2.5-fold increase in SCAR protein (*Figure 3B*). We verified these results by immunoblotting whole cell extracts prepared from the brains of 3$^{rd}$ instar larvae bearing homozygous mutations for *Atxn7* (*Ataxin7[KG02020]*), a loss-of-function allele (*Mohan et al., 2014a*) or *non-stop* (*not[02069]*), also a loss of function allele (*Poeck et al., 2001*). Larval brain extracts from *Atxn7* mutants showed about 2.5-fold increase in SCAR protein levels (*Figure 3C*). In *non-stop* mutant extracts SCAR was decreased approximately 75% (*Figure 3C*).

SAGA was first identified as a chromatin modifying complex and predominantly characterized as a transcriptional coactivator. Therefore, we examined whether Non-stop influences *SCAR* gene expression. In genome wide analysis of glial nuclei isolated from mutant larvae, *SCAR* transcript was significantly down regulated by one fold in Non-stop deficient glia, but not in *Sgf11* deficient glia (*Ma et al., 2016*). Since Sgf11 is required for Non-stop enzymatic activity, it is not clear what mechanism links Non-stop loss to reduction in *SCAR* gene expression. To confirm the relationship between Non-stop and *SCAR* gene expression, we used qRT-PCR to measure *SCAR* transcript levels in BG3 cells knocked-down for *non-stop*. We found a similar one half decrease in *SCAR* transcript levels (*Figure 3D*). However, examination of *Atxn7* knockdown showed no decrease in *SCAR* transcript levels (*Figure 3D*). Therefore, while *non-stop* loss leads to modest decreases in *SCAR* transcripts, *SCAR* gene expression is independent of *Atxn7* or *Sgf11*, suggesting this gene is not SAGA-dependent or dependent on Non-stop enzymatic activity. To determine the consequences of reducing *SCAR* transcripts by half on SCAR protein levels, we examined *SCAR* heterozygous mutant larval brains

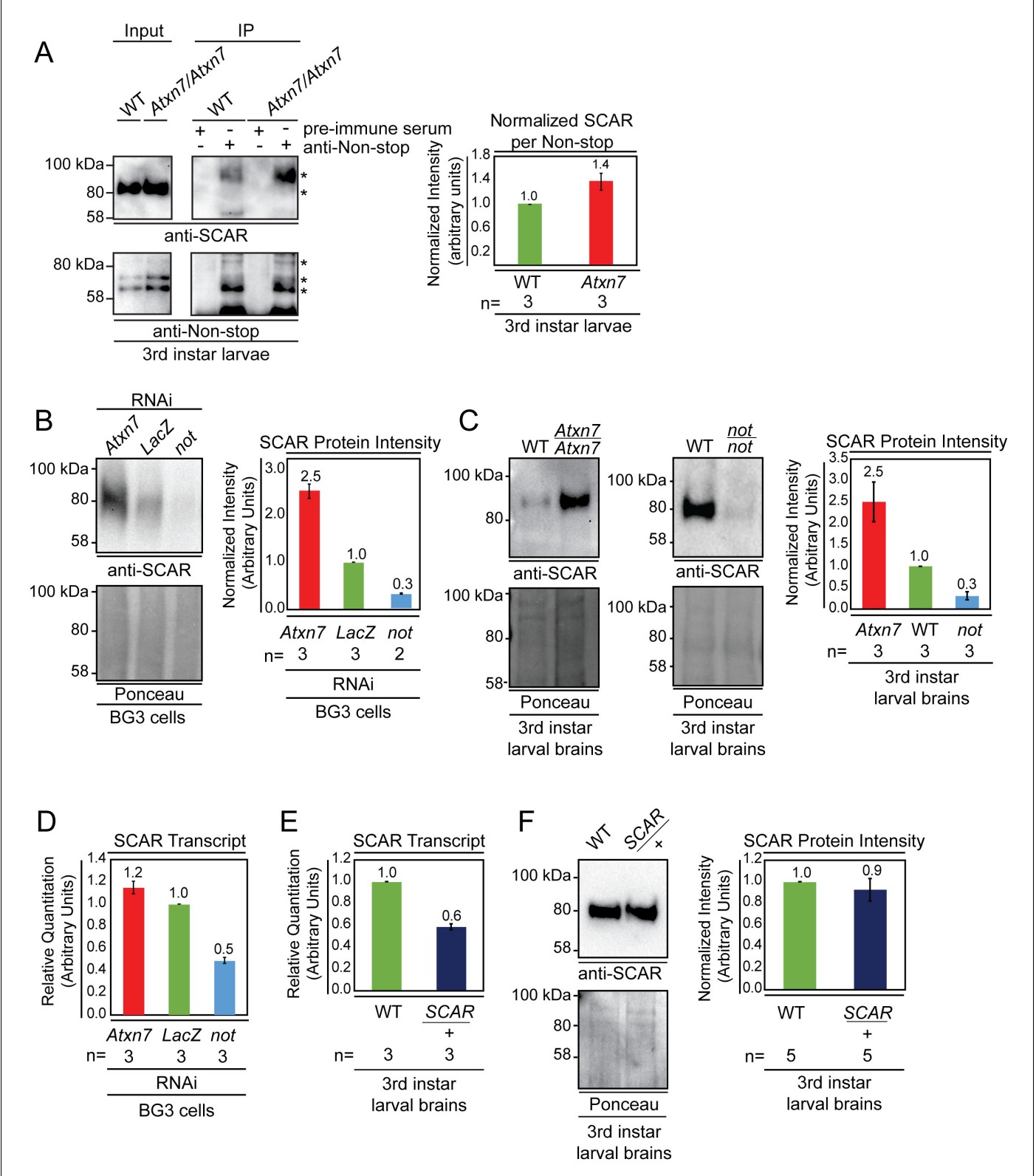

**Figure 3.** Non-stop and Atxn7 regulate SCAR protein levels. (**A**) Endogenous pull-down reveals increased interaction between Non-stop and SCAR in the absence of Atxn7. Whole cell extracts prepared from either OregonR (WT) or homozygous mutant *Ataxin7[KG02020]* (*Atxn7*) third instar larvae were subject to immunoprecipitation with pre-immune guinea pig serum or anti-Non-stop antibody as indicated. Immunoprecipitates were analyzed by immunoblotting with anti-Non-stop to verify capture, and with anti-SCAR antibody to assess interaction. To compare the relative proportion of SCAR

*Figure 3 continued on next page*

*Figure 3 continued*

interacting with Non-stop, immunoblotting signals were quantified by densitometry and the amount of SCAR interacting per given amount of Non-stop was calculated and expressed relative to WT control. Error bars represent standard error. (B and C) Loss of Atxn7 increases, and loss of Non-stop decreases SCAR protein levels. (B) BG3 cells were treated with dsRNA to knock down either *Atxn7*, *LacZ* (control), or *non-stop* (*not*). Denaturing whole cell extracts were analyzed by immunoblotting to determine SCAR protein levels. Total protein (Ponceau, lower panel) was used as loading control. SCAR intensity was quantified and values were normalized to Ponceau and expressed relative to *LacZ* control (right panel). Error bars represent standard error. (C) Brain and central nervous systems from WT, *non-stop*[02069] (*not*), or *Ataxin7*[KG02020] (*Atxn7*) third instar larvae were isolated and denaturing whole cell extracts prepared. SCAR protein levels were determined by immunoblotting and quantified as in B. (D, E, F) Atxn7- or Non-stop-mediated gene regulation are not sufficient to change SCAR protein levels. (D) Relative quantification of SCAR transcripts in BG3 cells treated with dsRNA targeting either *Atxn7*, *LacZ* (control), or *not*. Transcripts from *not* and *Atxn7* treated cells are expressed relative to *LacZ*. (E) Relative quantification of *SCAR* transcript levels in larval brains dissected from WT or *SCAR*[Delta37]/cyo-gfp (*SCAR*/+). *SCAR* transcript levels are set relative to WT. (F) Immunoblot for SCAR protein in larval brains dissected from OregonR (WT) or *SCAR*[Delta37]/cyo-gfp (*SCAR*/+). SCAR protein levels were quantified as in B.

DOI: https://doi.org/10.7554/eLife.49677.005

(*SCAR*[Delta37]/+). qRT-PCR analysis showed heterozygous *SCAR* brains produced 40% less *SCAR* transcript than wild-type brains (*Figure 3E*). However, heterozygous *SCAR* mutant and wild-type brains showed similar amounts of SCAR protein, demonstrating this amount of transcript is sufficient for full protein production (*Figure 3F*). Since *non-stop* mutants showed a similarly modest decrease in *SCAR* gene expression but a drastic decrease in protein levels, we tested the possibility that Non-stop counters SCAR protein degradation post-translationally.

Purification and mass spectrometry identified WRC members SCAR, HEM, and SRA-1, as interaction partners of the independent DUBm (*Figure 2A*). In *Drosophila*, HEM, SRA-1, and Abi recruit an unidentified deubiquitinase to balance SCAR regulation through a constant ubiquitination-proteasomal degradation mechanism facilitating rapid changes in SCAR protein amount and localization (*Chen et al., 2014*; *Kunda et al., 2003*). In mammals, the USP7 deubiquitinase acts as a molecular rheostat controlling proteasomal degradation of WASH (*Hao et al., 2015*).

To probe whether ubiquitinated SCAR is a substrate for Non-stop, the substrate binding pocket of Non-stop was modified to either augment (Non-stop(C406A)), or reduce (Non-stop(C406S)) substrate binding (*Morrow et al., 2018*). These mutants were immunoprecipitated from whole cell extracts prepared from transiently transfected BG3 cells. Immunoblotting revealed that Non-stop (C406A) bound to SCAR approximately 4-fold more than wild-type Non-stop while Non-stop(C406S) bound only about 1/3rd as well (*Figure 4A and B*, upper panels). To verify both mutants incorporated into SAGA, interaction with Ada2b was confirmed (*Figure 4A and B*, lower panels).

Entry into the proteasome for destruction can be triggered by polyubiquitination (*Grice and Nathan, 2016*; *Meulmeester et al., 2005*). To determine whether Non-stop counters polyubiquitination of SCAR, HA-ubiquitin and SCAR-FLAG were expressed in BG3 cells and either *LacZ* (negative control) or *non-stop* were knocked down followed by a brief period of proteasomal inhibition to preserve polyubiquitinated SCAR which would otherwise be rapidly destroyed by the proteasome (*Xia et al., 2012*). Immunoprecipitation of SCAR followed by immunoblotting with a ubiquitin-specific antibody revealed that knock-down of Non-stop nearly tripled levels of SCAR polyubiquitination (*Figure 4C*).

To determine whether Non-stop regulates SCAR entry into a proteasomal degradation pathway 3rd instar larval brains were cultured *ex vivo* to permit pharmacological inhibition of the proteasome in defined genetic backgrounds (*Rabinovich et al., 2015*; *Prithviraj et al., 2012*). Wandering 3rd instar larval brains were isolated from *non-stop* homozygous mutant larvae and cultured for 24 hr in the presence of MG132 proteasome inhibitor. Treating *non-stop* brains with MG132 rescued SCAR protein amounts to levels found in wild type brains (*Figure 4D*). Together, these data show Non-stop counters polyubiquitination and entry of SCAR into the proteasomal degradation pathway.

## Non-stop regulates SCAR protein levels and localization

To examine whether increasing Non-stop led to increased SCAR, we expressed recombinant Non-stop in BG3 cells and utilized indirect immunofluorescence to observe the amount and location of endogenous SCAR protein. Upon expression of Non-stop-FH in BG3 cells, endogenous SCAR protein levels increased five-fold compared with mock transfected cells (*Figure 5A*). Utilizing anti-HA

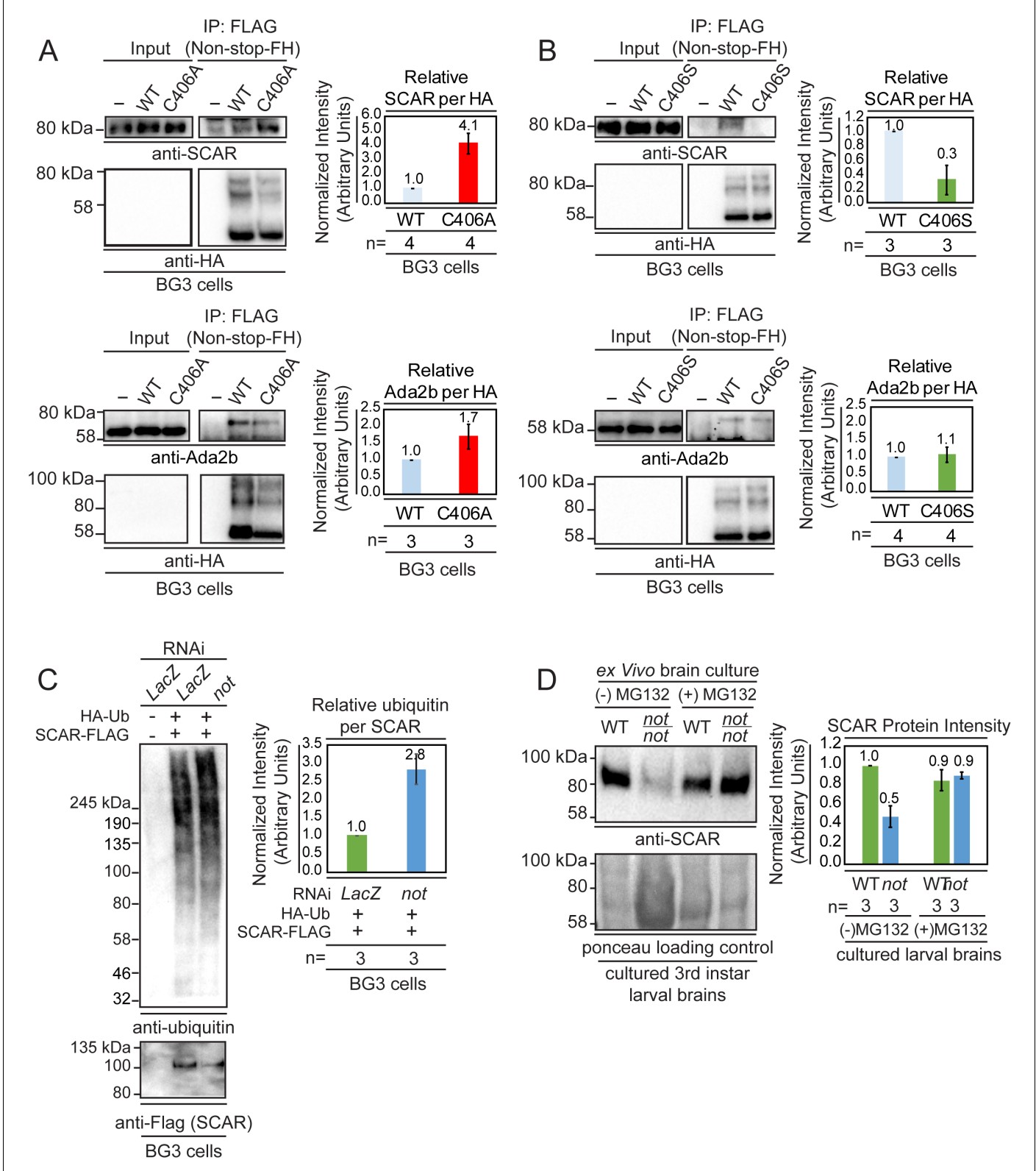

**Figure 4.** Non-stop binds ubiquitinated SCAR, regulates SCAR ubiquitination, and entry into the proteasome degradation pathway. (**A**) Non-stop catalytic mutation which increases substrate binding also increases interaction with SCAR. The Non-stop catalytic cysteine was mutated to alanine creating Non-stop C406A-2xFLAG-2xHA (C406A). The wild type version of Non-stop (WT), the mutant version of Non-stop (C406A), or no plasmid (-) were transfected into BG3 cells and whole cell extracts were prepared. Flag resin was used to immunoprecipitate Non-stop-FH protein. Extracts were

*Figure 4 continued on next page*

Figure 4 continued

immunoblotted for SCAR. Immunoblots for Ada2b control for incorporation into the SAGA complex. Immunoblots for HA verify the presence of the Non-stop-FH constructs. Ada2b and SCAR intensity were quantified and normalized to HA. Values are shown as relative to WT. Error bars represent standard error. (B) Non-stop catalytic mutation which decreases substrate binding also decreases interaction with SCAR. The Non-stop catalytic cysteine was mutated to serine (C406S). BG3 cells were mock transfected (-), transfected with Non-stop-FH, or with Non-stop C406S-FH (C406S) and whole cell extracts were prepared. Flag resin was used to immunoprecipitate the Non-stop-FH constructs. Pull-downs were immunoblotted for SCAR. Incorporation into SAGA was verified by Ada2b immunoblot. HA immunoblots verify the presence of the Non-stop-FH constructs. Ada2b and SCAR intensity was quantified and normalized to HA. Values are shown as relative to wild type. Error bars are standard error. (C) Non-stop counters polyubiquitination of SCAR. BG3 cells were treated with dsRNA targeting either *non-stop* or LacZ and transfected with HA-ubiquitin (HA-Ub) and SCAR-FLAG (SCAR-F). Control cells were treated with LacZ dsRNA and a mock transfection was performed. After six days, cells were treated with MG132 protease inhibitor for 6 hr and denaturing whole cell extracts were made. Anti-flag resin was used to capture SCAR-FLAG. Immunoblots were performed for ubiquitin (VU-1) and Flag to verify presence of SCAR. Protein intensity for VU-1 was measured and normalized to Flag protein intensity. Values are shown as relative to *LacZ*. Error bars represent standard error. (D) Non-stop counters proteasomal degradation of SCAR. WT or *not* brains were dissected from 3rd instar larva and cultured *ex vivo*. Half of the brains from each genotype were treated for 24 hr with protease inhibitor [50 μm MG132 (MG132+)] while the remaining brains served as untreated control (MG132-). Whole cell extracts were immunoblotted for SCAR. SCAR protein intensity was measured and normalized to Ponceau total protein control. Numbers are expressed as relative to WT untreated control. Error bars represent standard error.

DOI: https://doi.org/10.7554/eLife.49677.006

immunofluorescence to observe recombinant Non-stop, three distinct subcellular localization patterns were apparent: nuclear, cytoplasmic, and evenly distributed between nucleus and cytoplasm (*Figure 5A*). Increased SCAR protein co-compartmentalized with Non-stop, even within the nucleus, where SCAR function is more enigmatic (*Figure 5A*). To quantify co-compartmentalization of SCAR and Non-stop, the nuclear to cytoplasmic ratios of Non-stop-FH and endogenous SCAR were calculated by dividing immunofluorescence intensity within each nucleus by that of the corresponding cytoplasm. This was plotted as the ratio nuclear:cytoplasmic SCAR on the Y axis and the ratio nuclear:cytoplasmic Non-stop on the X axis. As the amount of Non-stop in the nucleus increased, so did the amount of SCAR in the nucleus, showing Non-stop was driving SCAR localization ($R^2 = 0.8$) (*Figure 5A*).

SCAR functions, within WRC, to activate Arp2/3 and facilitate branching of F-actin filaments (*Kurisu and Takenawa, 2009*). With correct spatiotemporal regulation of these actin-branching complexes, cells are able to establish appropriate shape and size through extension of actin-based protrusions (*Blanchoin et al., 2014*). BG3 cells have a characteristic bimodal appearance when plated in the absence of concanavalin A (*Liu et al., 2009*). If Non-stop overexpression were disrupting SCAR function through augmenting and mislocalizing SCAR, changes in cell shape and ability to form protrusions were predicted (*Machesky and Insall, 1998*). Using phalloidin to observe F-actin in cells overexpressing Non-stop, a substantial change in cell morphology was observed. Cells displayed fewer projections and covered less total area (*Figure 5B*). This was specific for cells expressing Non-stop in the cytoplasm or both nucleus and cytoplasm. Increased Non-stop signal associated closely with increased phalloidin signal. Overexpression of SCAR alone similarly led to reductions in cell area and number of projections observed (*Figure 5B*).

## Non-stop bears multiple, conserved, WRC interacting receptor sequence (WIRS) motifs

The basis for interaction between the SAGA DUBm, Arp2/3, and WRC was sought. WRC subunits HEM, SRA-1, and Abi mediate interaction with an unidentified deubiquitinase to counteract proteasomal degradation of SCAR (*Kunda et al., 2003*). WRC subunits Abi and SRA-1 mediate interaction with WRC interacting receptor sequences (WIRS)-bearing proteins which contribute to spatiotemporal regulation of SCAR (*Chen et al., 2014*). Recall, Hem and Sra-1 were present in mass spectrometry analysis of Group 2 proteins, suggesting Non-stop was in vicinity of this surface of WRC.

Primary sequence analysis of DUBm subunits revealed four WIRS-conforming sequences on Non-stop and three on mammalian Non-stop orthologue USP22. These were numbered from 0 to 3, with WIRS1 and 3 conserved in location between *Drosophila* and mammalian Non-stop. (*Figure 6A*). None of the other DUBm components bore conserved WIRS motifs. Although analysis of the yeast orthologue of Non-stop (UBP8) did not reveal a WIRS-like motif, the Atxn7 orthologue Sgf73 did. In

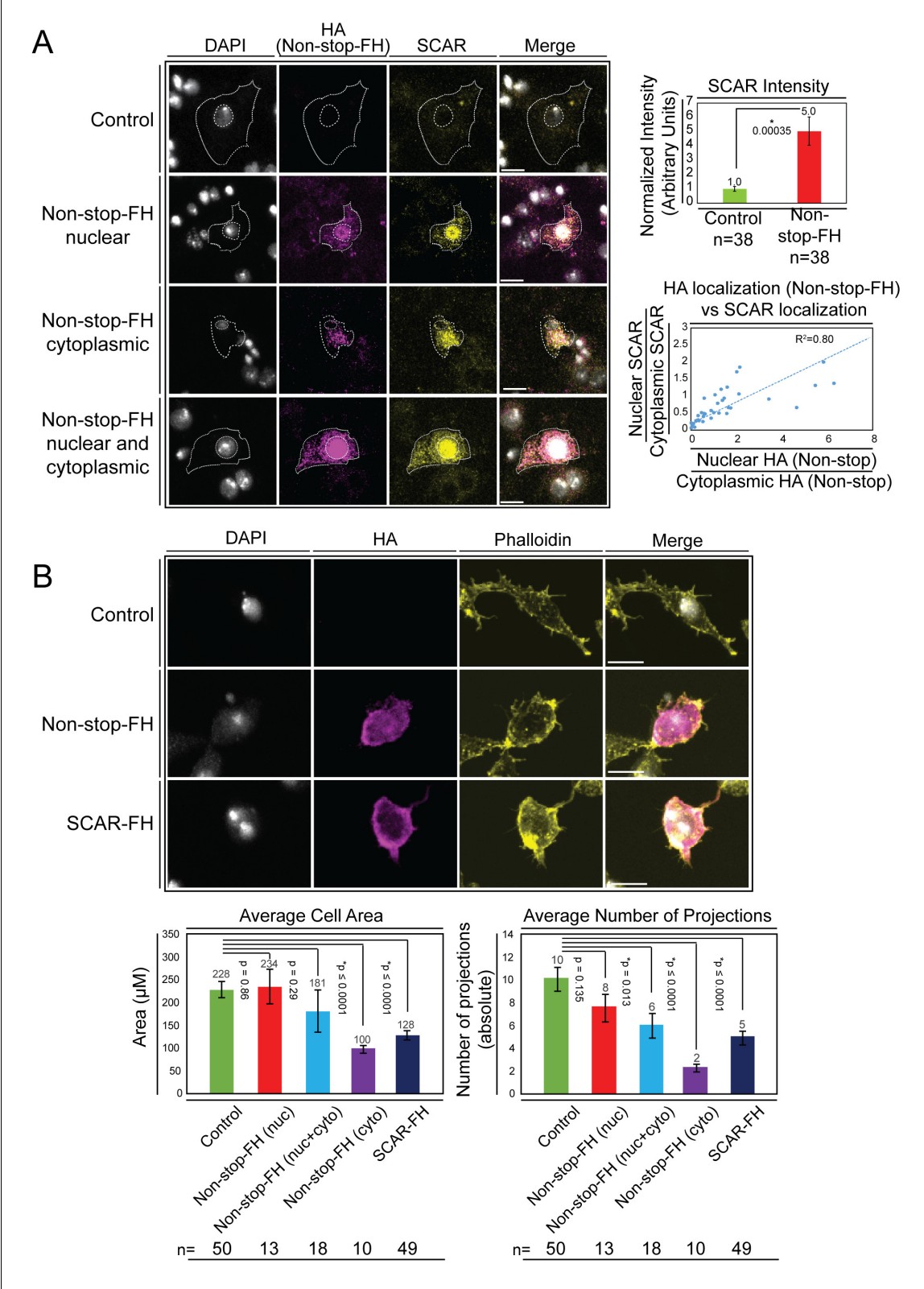

**Figure 5.** Overexpression of Non-stop increases SCAR levels, directs SCAR localization, and alters cell morphology – reducing cell area and number of cell protrusions. (**A**) Augmenting Non-stop increases local levels of SCAR. *Drosophila* BG3 central nervous system cells were transfected with no plasmid (control) or with Non-stop-FH expression vector as indicated. Exogenous Non-stop and endogenous SCAR were located and quantified through nuclear localization by DAPI (DNA) (white) and indirect immunofluorescence toward: Non-stop-FH (anti-HA) (magenta), and anti-SCAR (yellow).

*Figure 5 continued on next page*

*Figure 5 continued*

Scale bar is 10 μM. Hashed lines outline DAPI (inner circle) to approximate nuclear location and the outermost SCAR signal to approximate the cell edge (outer circles). Three categories of HA (Non-stop-FH) expression are shown: Non-stop 'nuclear', Non-stop 'cytoplasmic', and Non-stop 'nuclear and cytoplasmic.' Total SCAR signal was measured in HA positive cells (Non-stop-FH) and mock transfected cells (Control). The average increase in endogenous SCAR immunofluorescence intensity was determined relative to control (top, right panel). Error bars are standard error. To examine the relationship between Non-stop localization and increased SCAR protein, SCAR immunofluorescence intensity was measured in the nucleus (as defined by DAPI) and in the cytoplasm. A ratio of nuclear SCAR intensity divided by cytoplasmic SCAR intensity was calculated. HA intensity was similarly measured in the nucleus and cytoplasm. A ratio of nuclear HA intensity to cytoplasmic HA intensity was calculated. The HA ratio was plotted on the X axis and the SCAR ratio was plotted on the Y axis. A trend line was calculated and the $R^2$ was determined to be 0.80. (**B**) Increasing Non-stop alters F-actin organization similarly to increasing SCAR – reducing cell area and number of cell protrusions. BG3 cells were transfected with no-plasmid (Control), Non-stop-FH, or SCAR-FH. The location of exogenously expressed proteins were determined as above and F-actin was visualized using phalloidin. Cell area and total number of cellular projections were counted by quantifying the phalloidin signal distribution. HA-positive cells (Non-stop-FH or SCAR-FH) were compared to mock transfected (Control) cells (bottom panels). Cells expressing Non-stop-FH were split into three categories of HA (Non-stop-FH) expression as above. T-tests were used to compare samples and p-values are shown. Error bars are standard error.

DOI: https://doi.org/10.7554/eLife.49677.007

yeast, the DUBm can also be found separate from SAGA, although not without Sgf73 (*Lim et al., 2013*). Therefore, we focused on Non-stop WIRS motifs.

To verify Non-stop WIRS motifs are important for Non-stop/SCAR interaction, putative WIRS motifs were inactivated using a phenylalanine to alanine substitution, previously demonstrated to be effective at inactivating these motifs (*Chen et al., 2014*). This mutant displayed an approximate 80% reduction in SCAR binding, but still integrated into SAGA as observed by maintenance of Atxn7 binding (*Figure 6B*).

To determine whether Non-stop WIRS sequences are necessary for Non-stop-mediated modulation of SCAR function, a series of phenylalanine to alanine point mutants of individual WIRS domains (WIRS F-A) were generated and expressed in BG3 cells. As above (*Figure 5*), the ability of Non-stop WIRS mutants to increase the levels of endogenous SCAR protein, to change cell area, and to change the number of protrusions per cell was measured. Immunofluorescence was used to detect the HA epitope on exogenously expressed Non-stop and an antibody toward endogenous SCAR was used to observe SCAR protein levels. Cells expressing moderate amounts of Non-stop as judged by the spectrum of fluorescence intensity were analyzed to avoid assaying cells with either too little or too much exogenous Non-stop relative to endogenous protein. The ratio of HA fluorescence to SCAR fluorescence in each mutant was calculated and compared to wild-type Non-stop (*Figure 6C*). In each case, the Non-stop mutant produced less SCAR protein than wild type, suggesting that these motifs are indeed important for Non-stop mediated increases in SCAR protein levels (*Figure 6C*). Mutant Non-stop was also less potent for decreasing cell area and decreasing the number of cell protrusions (*Figure 6D*). The effect of mutating each WIRS was not equivalent, suggesting a hierarchy for which is most important for WRC stabilization.

In *Drosophila*, specific cell-type and subcellular location of actin branching enzymes facilitate fine tuning of actin remodeling in space and time (*Zallen et al., 2002*). SCAR and the single *Drosophila* WASp homolog act non-redundantly to orchestrate actin branching in different tissues and to regulate different developmental decisions. SCAR is particularly important in axon development in the central nervous system and adult eye morphology, but also essential for blastoderm organization and egg chamber structure during development. Within these tissues, cells display both cytoplasmic and nuclear localization of SCAR (*Zallen et al., 2002*).

Considering the demonstrated importance of Atxn7, Non-stop, and SCAR in the nervous system and eye, the interplay between these genes was examined in 3rd instar larval brains. To observe an output of SCAR function, phalloidin was used to visualize F-actin. In Non-stop mutants, F-actin intensity was reduced by approximately 70% (*Figure 7A*). In Atxn7 mutants, F-actin levels increased approximately 1.7 times (*Figure 7B*). To test for genetic interaction between the SAGA DUBm and SCAR, heterozygous mutations of Atxn7 and SCAR were combined and F-actin levels determined (*Figure 7C*). In Atxn7 heterozygotes, F-actin levels were not significantly different than wild-type. In SCAR heterozygotes, however, F-actin levels were decreased. Combining Atxn7 and SCAR heterozygous mutants rescued F-actin levels. Together, these results support a model where Atxn7 restrains Non-stop from stabilizing and increasing levels of SCAR, which then increases F-actin.

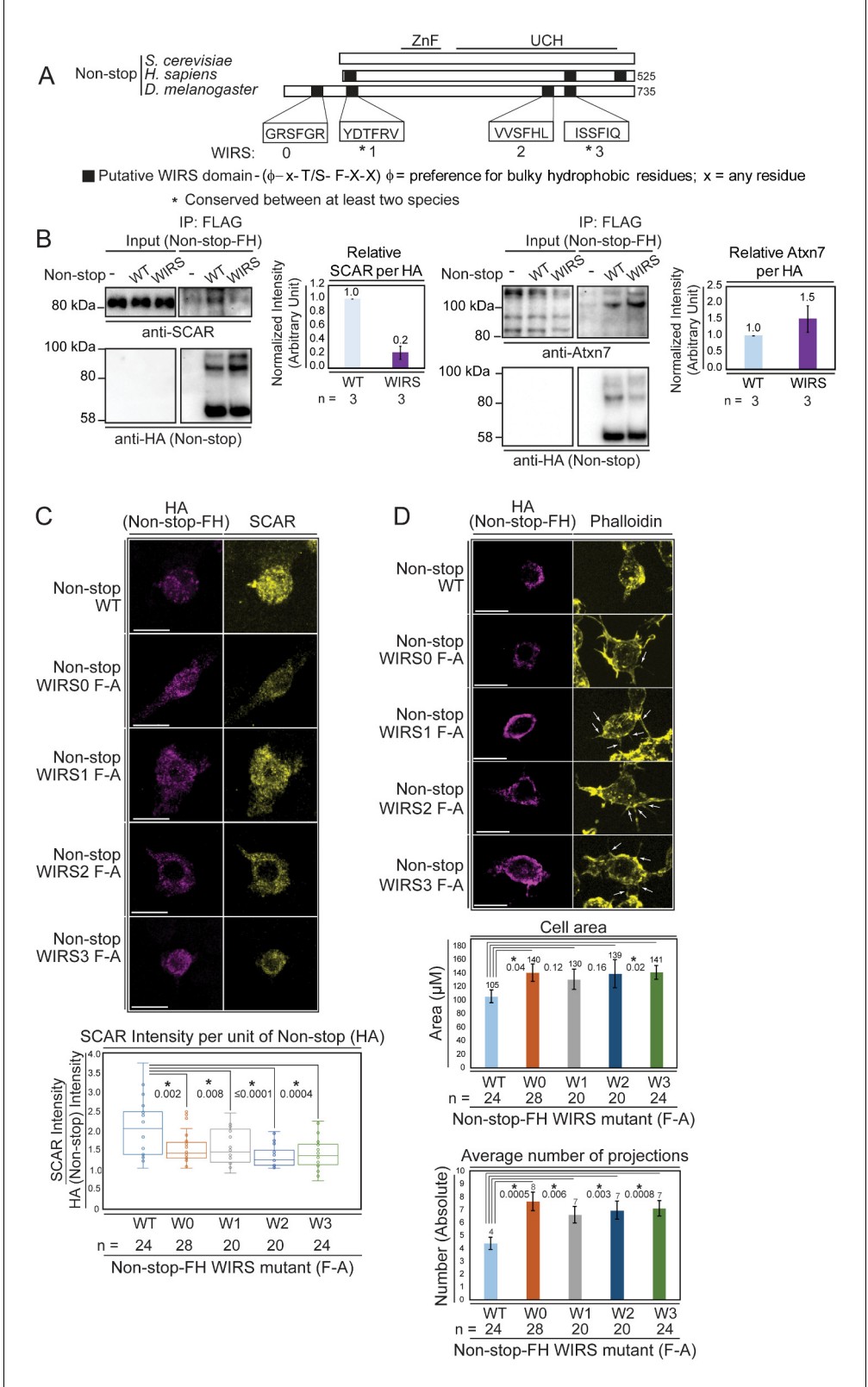

**Figure 6.** Non-stop bears conserved WIRS motifs. Mutating these reduces Non-stop's ability to bind SCAR, increase SCAR levels, reduce cell area, and reduce number of cell protrusions. (**A**) Alignment of Non-stop protein sequences reveals a conserved distribution of WIRS motifs between higher eukaryotes. (**B**) A WIRS mutant version of Non-stop-FH was created bearing phenylalanine to alanine point mutations in all four WIRS domains (WIRS). BG3 cells were mock transfected, transfected with Non-stop-FH (WT), or Non-stop WIRS FA-FH (WIRS). Whole cell extracts were prepared and

*Figure 6 continued on next page*

*Figure 6 continued*
immunoprecipitated using anti-FLAG resin to capture exogenous Non-stop and immunoblotted for SCAR. Immunoblot for Atxn7 verified incorporation into the SAGA complex. Immunoblot for HA verified capture of Non-stop-FH. Protein intensity of Atxn7 and SCAR were normalized to HA intensity. Intensities are shown relative to WT Non-stop. Error bars are standard error. (C) BG3 cells were transfected with either wild type (WT) or Non-stop-FH harboring a point mutation in one of the four putative WIRS motifs (W0–W3) as indicated. Exogenously expressed Non-stop and endogenous SCAR were detected by indirect immunofluorescence. Scale bar is 10 µm. Total fluorescence intensity of SCAR and HA were measured in HA containing cells. Box plots show the SCAR intensity divided by the intensity of HA. T-tests were used to compare samples and p-values are shown. P-values marked with an asterisk are significant. (D) BG3 cells transfected with either wild-type or Non-stop-FH harboring a point mutation in one of the four putative WIRS motifs (W0–W3) were immunostained for HA (magenta) and phalloidin (yellow). Scale bar is 10 µm. Arrows point to localized accumulation of Non-stop WIRS mutant protein, which coincide with aberrant actin protrusions commonly observed upon WRC or Arp2/3 loss of function. Cell area and number of projections protruding from cells was determined for HA positive cells. T-tests were used to compare the samples and p-values are shown. P-values marked with an asterisk are significant. Error bars are standard error.
DOI: https://doi.org/10.7554/eLife.49677.008

To test whether SCAR rescues an Atxn7 phenotype, retinal axon targeting in 3$^{rd}$ instar larval brains was observed using an anti-chaoptin antibody, a retinal axon-specific protein (*Figure 8*). Neural-specific expression of Atxn7 RNAi (*Dietzl et al., 2007*), driven by elav-Gal4, results in defective axon targeting. Similarly, knockdown of either Non-stop (*Perkins et al., 2015*) or SCAR (*Perkins et al., 2015*) led to defective axon targeting. Gcn5 knock-down (*Dietzl et al., 2007*), however, resulted in wild-type axon targeting. To test whether defective axon targeting upon Atxn7 loss, could be rescued by reducing SCAR (*Perkins et al., 2015*), Atxn7 was knocked-down in a SCAR heterozygous background. Although there were fewer defects in axon targeting (62% normal vs 52% normal), this difference was not sufficient to achieve statistical significance (p=0.73).

## Discussion

Purification of Atxn7-containing complexes indicated that Atxn7 functions predominantly as a member of SAGA (*Mohan et al., 2014a*). In yeast, the Atxn7 orthologue, Sgf73, can be separated from SAGA along with the deubiquitinase module by the proteasome regulatory particle (*Lim et al., 2013*). Without Sgf73, the yeast deubiquitinase module is inactive (*Samara et al., 2010*). In higher eukaryotes, Atxn7 increases, but is not necessary for Non-stop/USP22 enzymatic activity *in vitro* (*Mohan et al., 2014a*; *Lan et al., 2015*). In *Drosophila*, loss of Atxn7 leads to a Non-stop over activity phenotype, with reduced levels of ubiquitinated H2B observed (*Mohan et al., 2014a*). Here, purification of Non-stop revealed the active SAGA DUBm associates with multi-protein complexes including WRC and Arp2/3 complexes separate from SAGA. SCAR was previously described to be regulated by a constant ubiquitination/deubiquitination mechanism (*Kunda et al., 2003*). SCAR protein levels increased upon knockdown of *Atxn7* and decreased upon knockdown of *non-stop*. Decreases in SCAR protein levels in the absence of *non-stop* required a functional proteasome.

Conversely, overexpression of Non-stop in cells led to increased SCAR protein levels and this increased SCAR protein colocalized to subcellular compartments where Non-stop was found. Nuclear Arp2/3 and WRC have been linked to nuclear reprogramming during early development, immune system function, and general regulation of gene expression (*Yoo et al., 2007*; *Miyamoto et al., 2013*; *Taylor et al., 2010*). Distortions of nuclear shape alter chromatin domain location within the nucleus, resulting in changes in gene expression (*Navarro-Lérida et al., 2015*; *Rodriguez-Mesa et al., 2012*). Nuclear pore stability is compromised in SAGA DUBm mutants, resulting in deficient mRNA export (*Köhler et al., 2008*). Similarly, mutants of F-actin regulatory proteins, such as Wash, show nuclear pore loss (*Verboon et al., 2015*). Non-stop may contribute to nuclear pore stability and mRNA export through multiple mechanisms.

When we examined the basis for this unexpected regulatory mechanism, we uncovered a series of WIRS motifs (*Chen et al., 2014*) conserved in number and distribution between flies and mammals. These sequences functionally modulate Non-stop ability to increase SCAR protein levels. Point mutants of each WIRS resulted in less SCAR protein per increase in Non-stop protein. WIRS mutant Non-stop retained the ability to incorporate into SAGA, indicating these are separation of function mutants.

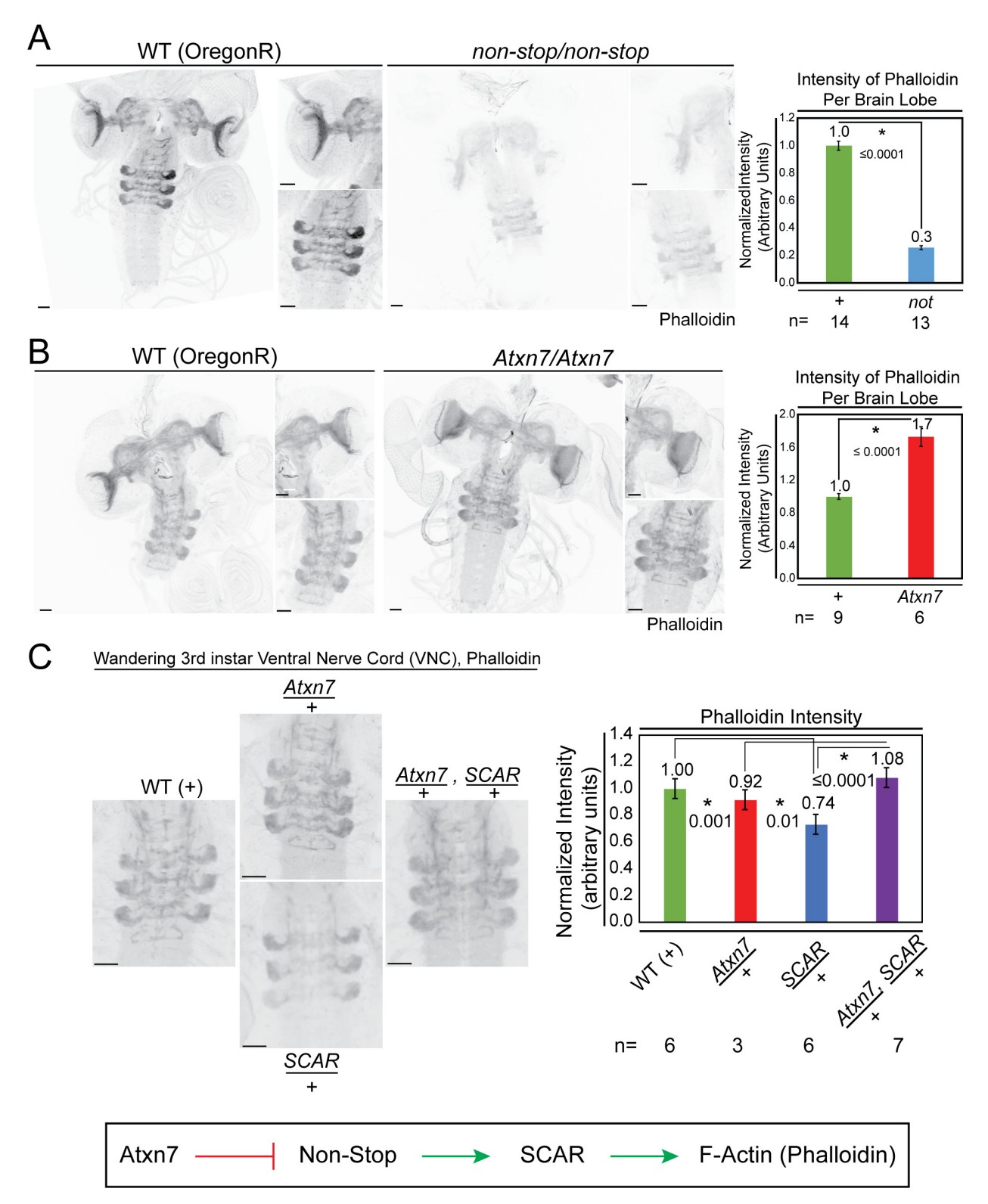

**Figure 7.** Atxn7 and Non-stop, act through the SCAR pathway in order to regulate the actin cytoskeleton *in vivo*. (A) Phalloidin staining in third instar larval brains dissected from *not02069* reveals a decrease in neural F-actin. (B) Conversely, phalloidin staining in *Atxn7KG02020* shows an increase in F-actin. Microscope acquisition settings were identical to allow comparison. Scale bar is 50 μM. Charts show the averaged phalloidin fluorescence intensity measurements for individual brain lobes. Wild type average intensity was set to one and mutants were normalized to wild -type. A t-test was used to

*Figure 7 continued on next page*

*Figure 7 continued*

compare the samples and the p-value is shown and asterisks indicate significance. Error bars are standard error. (C) Phalloidin staining in wild type, *Atxn7^{KG02020}/+, SCAR^{[Δ37]} /+*, and *Atxn7^{KG02020}, SCAR^{[Δ37]} /+, +* third instar VNC shows that *Atxn7* mutation can rescue the defects seen in *SCAR* heterozygotes. Scale bar is 20 μM. Fluorescence intensity was measured for each VNC. Wild-type average intensity was set to one and mutants were normalized to wild-type. Error bars are standard error. A t-test was used to compare the samples and significant values are listed with an asterisks and p-value. Line diagram outlines an explanation for these observations (bottom).

DOI: https://doi.org/10.7554/eLife.49677.009

Overall, these findings suggest that the cell maintains a pool of Non-stop that can be made available to act distally from the larger SAGA complex to modulate SCAR protein levels (*Figure 9*). In yeast, the proteasome regulatory particle removes the DUBm from SAGA (*Lim et al., 2013*). In higher eukaryotes, caspase-7 cleaves Atxn7 at residues which would be expected to release the DUBm, although this remains to be shown explicitly (*Young et al., 2007*). The mechanisms orchestrating entry and exit of the DUBm from SAGA remain to be explored.

# Materials and methods

## Key resources table

| Reagent type (species) or resource | Designation | Source or reference | Identifiers | Additional information |
|---|---|---|---|---|
| Gene (*Drosophila melanogaster*) | not | NA | FLYB: FBgn0013717 | |
| Gene (*D. melanogaster*) | Atxn7 | NA | FLYB: FBgn0031420 | |
| Gene (*D. melanogaster*) | SCAR | NA | FLYB: FBgn0041781 | |
| Gene (*D. melanogaster*) | Gcn5 | NA | FLYB: FBgn0020388 | |
| Gene (*D. melanogaster*) | Ada2b | NA | FLYB: FBgn0037555 | |
| Strain, strain background (*D. melanogaster*) | Elav-Gal4 | Bloomington Drosophila Stock Center | BDSC:458 RRID:BDSC_458 | Genotype:P(w[+mW.hs] =GawB)elav[C155] |
| Strain, strain background (*D. melanogaster*) | Uas-Gcn5 RNAi | Vienna Drosophila Resource Center | VDRC:21786 RRID:FlyBase_ FBst0454233 | Construct ID:11218 |
| Strain, strain background (*D. melanogaster*) | Uas-SCAR RNAi | Bloomington Drosophila Stock Center | BDSC:36121 RRID:BDSC_36121 | Genotype: y[1] sc[*] v[1]; P(y[+t7.7] v[+t1.8]=TRiP. HMS01536)attP40 |
| Strain, strain background (*D. melanogaster*) | Uas-Not RNAi | Bloomington Drosophila Stock Center | BDSC:28725 rebalanced with Tm6b, Tb RRID:BDSC_28725 | Genotype: y[1] v[1]; P(y[+t7.7] v[+t1.8]=TRiP. JF03152)attP2/TM6B, Tb |
| Strain, strain background (*D. melanogaster*) | Uas-Atxn7 RNAi | Vienna Drosophila Resource Center | VDRC:102078 RRID:FlyBase_FBst0473949 | Construct ID:110634 |
| Strain, strain background (*D. melanogaster*) | OreR | DGGR | Catalog number: 109612 RRID:DGGR_109612 | |
| Strain, strain background (*D. melanogaster*) | Non-stop[02069] | Bloomington Drosophila Stock Center | BDSC:11553 Rebalanced with Tm3 GFP RRID:BDSC_11553 | Genotype: P(ry[+t7.2]=PZ)not[02069] ry[506]/TM3, P(w[+mC]=GAL4 twi. G)2.3, P(UAS-2xEGFP) AH2.3, Sb[1] Ser[1] |

*Continued on next page*

Continued

| Reagent type (species) or resource | Designation | Source or reference | Identifiers | Additional information |
|---|---|---|---|---|
| Strain, strain background (*D. melanogaster*) | SCAR [Delta37] | Bloomington Drosophila Stock Center | BDSC:8754 Rebalanced with CyoGFP RRID:BDSC_8754 | Genotype: w[*]; SCAR[Delta37] P(ry[+t7.2]=neoFRT)40A/CyO,, P(w[+mC]=GAL4 twi.G)2.2, P(UAS-2xEGFP)AH2.2. Cross BDSC:6662 and BDSC:8754 |
| Strain, strain background (*D. melanogaster*) | Atxn7[KG02020] | Bloomington Drosophila Stock Center | BDSC: 14255 Rebalanced with CyoGFP RRID:BDSC_14255 | Genotype: y[1] w[67c23]; P(y[+mDint2] w[BR.E.BR]=SUPor P) CG9866[KG02020]/CyO, P(w[+mC]=GAL4 twi.G)2.2, P(UAS-2xEGFP)AH2.2. |
| Strain, strain background (*D. melanogaster*) | Uas-Atxn7 RNAi, SCARΔ37 | This Paper | | Materials and methods Subsection Fly Strains Genotype: w[*]; SCAR[Delta37] P(ry[+t7.2]=neoFRT) 40A/CyO, P(w[+mC]=GAL4 twi.G)2.2, P(UAS-2xEGFP)AH2.2.; P{KK110634} VIE-260B/TM6C, cu[1] Sb[1] Tb[1] |
| Cell line (*D. melanogaster*) | ML-DmBG3-c2 | Drosophila Genomics Resource Center #68 | FLYB: FBtc0000068; RRID:CVCL_Z728 | FlyBase symbol: ML-DmBG3-c2 |
| Cell line (*D. melanogaster*) | S2-DRSC | Drosophila Genomics Resource Center #181 | FLYB: FBtc0000181; RRID:CVCL_Z992 | FlyBase symbol: S2-DRSC |
| Antibody | Guinea Pig anti Non-stop | (*Mohan et al., 2014a*) | | Western Blot Dilution (1:1000) IF Dilution (1:150) |
| Antibody | Mouse anti Chaoptin (Monoclonal) | Developmental Studies Hybridoma Bank | 24B10 RRID:AB_528161 | IF Dilution (1:250) |
| Antibody | Goat anti Rat IGG −568 (Polyclonal) | Invitrogen | A-11077 RRID:AB_141874 | IF Dilution (1:1000) |
| Antibody | Goat anti Mouse IGG-488 (Polyclonal) | Invitrogen | A-11001 RRID:AB_2534069 | IF Dilution (1:1000) |
| Antibody | Goat anti Guinea pig IGG- 488 (Polyclonal) | Invitrogen | A11073 RRID:AB_142018 | IF Dilution (1:1000) |
| Other | Phalloidin-488 | Invitrogen | A12379 | IF Dilution (1:20) |
| Other | Phalloidin-568 | Invitrogen | A12380 RRID:AB_2810839 | IF Dilution (1:20) |
| Other | Vecta Shield | Vector Labs | H-1200 | |
| Antibody | Rat anti HA-HRP (Monoclonal) | Roche | 12013819001 RRID:AB_390917 | Western Blot Dilution (1:500) |
| Antibody | Mouse anti SCAR (Monoclonal) | Developmental Studies Hybridoma Bank | P1C1-SCAR RRID:AB_2618386 | Western Blot Dilution (1:250) IF Dilution (1:100) |
| Antibody | Rabbit anti Atxn7 | (*Mohan et al., 2014a*) | | Western Blot Dilution (1:2000) |
| Antibody | Guinea Pig anti Ada2b | Gift from Jerry L Workman (*Kusch et al., 2003*) | | Western Blot Dilution (1:1000) |
| Antibody | Goat anti Guinea Pig HRP (Polyclonal) | Jackson Immuno Research INC | 106-035-003, RRID:AB_2337402 | Western Blot Dilution (1:10000) |

*Continued*

| Reagent type (species) or resource | Designation | Source or reference | Identifiers | Additional information |
|---|---|---|---|---|
| Antibody | Goat anti mouse HRP (Polyclonal) | Jackson Immuno Research INC | 115-035-003, RRID:AB_10015289 | Western Blot Dilution (1:5000) |
| Antibody | Goat anti Rabbit HRP (Polyclonal) | Jackson Immuno Research INC | 111-035-003 RRID:AB_2313567 | Western Blot Dilution (1:10000) |
| Antibody | Rat anti HA (Monoclonal) | Roche | 11867423001 RRID:AB_390918 | IF Dilution (1:250) |
| Recombinant DNA reagent | pMT-HA-Ub | gift from Jianhang Jia | | gift from Jianhang Jia |
| Recombinant DNA reagent | Scar-FH | Berkley Expression Clone Collection | FMO14142 | |
| Recombinant DNA reagent | Scar-Flag | This paper | | Materials and methods Subsection Plasmids Quick change on FMO14142 F Primer:GATGAC GACAAGGTCA AACTTGCTGCTTAGACT AGTTCTAGT R Primer: ACTAGAACTAGTC TAAGCAGCAA GTTTGACCTTGTCGTCATC |
| Recombinant DNA reagent | Prmha3-Non-stop-2XFlag-2XHA | This Paper | Sequence ID: AAD53181.1 | Materials and methods Subsection Plasmids |
| Recombinant DNA reagent | Prmha3-Non-stop-2XFlag-2XHA-0FA | This Paper | | Materials and methods Subsection Plasmids Quick change mutagenesis on Prmha3-Non-stop-2XFlag-2XHA WIRS0 phenylalanine to alanine F: GCAGTGGCCGAAGC GCCGGCAGGGGAACGGAAC GGTGGGC WIRS0 phenylalanine to alanine R: CCGTTCCCCTGCCGGCGC TTCGGCCACTGCTGCTGCTGC |
| Recombinant DNA reagent | Prmha3-Non-stop-2XFlag-2XHA-1FA | This Paper | | Materials and methods Subsection Plasmids Quick change mutagenesis on Prmha3-Non-stop-2XFlag-2XHA WIRS1 phenylalanine to alanine F: CAGCTACGATACAGCCC GGGTCAT CGACGCCTACTTCGCTGCTTGCG WIRS1 phenylalanine to alanine R: GGCGTCGATGACCCGGG CTGTATCGT AGCTGTGCTCCTTCACATAGC |
| Recombinant DNA reagent | Prmha3-Non-stop-2XFlag-2XHA-2FA | This Paper | | Materials and methods Subsection Plasmids Quick change mutagenesis on Prmha3-Non-stop-2XFlag-2XHA WIRS2 phenylalanine to alanine F: CCAGCGTGGTGT CGGCCCATTTG AAACGCTTCGAGCACTCAGCTCTG WIRS2 phenylalanine to alanine R: CGAAGCGTTTCA AATGGGCCGAC ACCACGCTGGGCAGAGTGCGCAG |

*Continued on next page*

Continued

| Reagent type (species) or resource | Designation | Source or reference | Identifiers | Additional information |
|---|---|---|---|---|
| Recombinant DNA reagent | Prmha3-Non-stop-2XFlag-2XHA-3FA | This Paper | | Materials and methods Subsection Plasmids Quick change mutagenesis on Prmha3-Non-stop-2XFlag-2XHA WIRS3 phenylalanine to alanine F: CGCAAGATCTCCTCGGCCATTCA ATTCCCCGTGGAGTTCGACATG WIRS3 phenylalanine to alanine R: CCACGGGGAAT TGAATGGCCGAGG AGATCTTGCGATCGATCAGAGC |
| Recombinant DNA reagent | Prmha3-Non-stop-2XFlag-2XHA-WIRSFA | This Paper | | Materials and methods Subsection Plasmids Quick change mutagenesis on Prmha3-Non-stop-2XFlag-2XHA all of the above primers sequentially |
| Recombinant DNA reagent | Prmha3-Non-stop-2XFlag-2XHA-C406A | This Paper | | Materials and methods Subsection Plasmids Quick change mutagenesis on Prmha3-Non-stop-2XFlag-2XHA Non-stop C406A F: CTTAATCTGGGCGCCACTG CCTTCATGAACTGCATCGTC Non-stop C406A R: GACGATGCAGTTCATGAAG GCAGTGGCGCCCAGATTAAG |
| Recombinant DNA reagent | Prmha3-Non-stop-2XFlag-2XHA-C406S | This Paper | | Materials and methods Subsection Plasmids Quick change mutagenesis on Prmha3-Non-stop-2XFlag-2XHA Non-stop C406S F: CTTAATCTGGGCGCC ACTAGCTTCATGAACTGCATCGTC Non-stop C406S R: GACGATGC AGTTCATGAAGCTAGTG GCGCCCAGATTAAG |
| Sequence-based reagent | dsRNA Lacz | | | dsRNA-LacZ-R: GCTAATACGACTCAC TATAGGCCAAACATGACCAR GATTACGCCAAGCT dsRNA-LacZ-F: GCTAATACGACTCACTATAG GCCAAACGTCCCATTCGCC ATTCAGGC |
| Sequence-based reagent | dsRNA not | http://www.flyrnai.org | DRSC11378 | Primer F: TAATACGACTCAC TATAGGCGCAGGCTGAACTGTTTG Primer R: TAATACGACTCACTATAGGT CTATTCCGGCTCCCGTT |
| Sequence-based reagent | dsRNA not | http://www.flyrnai.org | BKN21994 | Primer F: TAATACGACTCACTAT AGGACTTGACCCACGTGTCCTTC Primer R:TAATACGACTCACT ATAGGATTG ACCAGATCTTCACGGG |
| Sequence-based reagent | dsRNA Atxn7 | http://www.flyrnai.org | DRSC35628 | Primer F:TAATACGACTCACT ATAGGCGACATGGAAAAGGTCATCA Primer R:TAATACGACTCAC TATAGGG GAAACCTGCCTTCGTGTAA |

*Continued on next page*

*Continued*

| Reagent type (species) or resource | Designation | Source or reference | Identifiers | Additional information |
|---|---|---|---|---|
| Sequence-based reagent | dsRNA Atxn7 | http://www.flyrnai.org | DRSC23138 | Primer F: TAATACGAC TCACTATAGGCTGTTA AGCTGGAGGCCAAGPrimer R: TAATACGACTCACTATAGGG CCCTCTTATTGCACCTCAG |
| Sequence-based reagent | Taqman Assay: not | Thermofisher | TaqManID: Dm01823071_g1 | |
| Sequence-based reagent | Taqman Assay: Atxn7 | Thermofisher | TaqManID: Dm01800874_g1 | |
| Sequence-based reagent | Taqman Assay: SCAR | Thermofisher | TaqManID: Dm01810606_g1 | |
| Sequence-based reagent | Taqman Assay: RPL32 | Thermofisher | TaqMan ID: Dm02151827_g1 | |
| Commercial assay or kit | high capacity cDNA reverse transcription kit | ThermoFisher | 4374966 | |
| Commercial assay or kit | TaqMan Universal PCR Master Mix | ThermoFisher | 4364340 | |
| Chemical compound, drug | MG132 | Sigma-Aldrich | C2211 | |
| Software, algorithm | ImageJ | https://imagej. nih.gov/ij/ | | |

## Cell culture

ML-DmBG3-c2 (Drosophila Genomics Resource Center #68, RRID:CVCL_Z728, FBtc0000068) cells were grown in Schneider's media supplemented with 10% fetal bovine serum and 1:10,000 insulin. S2_DRSC cells (Drosophila Genomics Resource Center #181, RRID:CVCL_Z992) were maintained in Schneider's media supplemented with 10% fetal bovine serum and 1% penicillin-streptomycin (ThermoFisher, Catalog number: 15070063, 5000 U/ml). BG3 and S2 cell lines were ordered fresh from the Drosophila Genomics Resource Center and verified free from mycoplasma by PCR test or by microscopy. They were further verified by characteristics including survival in growth medium (these cell lines grow in distinct media – one cannot survive in the wrong media), their distinct morphology, and cell doubling time.

## Transfection

BG3 cells were plated at $1 \times 10^6$ cells/ml, 24 hr later they were transfected using Lipofectamine 3000 (ThermoFisher catalog number: L3000015) following manufacturer's directions. The Lipofectamine 3000 was used at 1 µl per µg of DNA and p3000 reagent was used at 1 µl per µg of DNA. To induce expression of proteins from plasmids containing a metallothionein promoter 250 µm of $CuSO_4$ was added to the media. For transfections taking place in six-well dishes 1 µg of DNA was used per well, 4 µg of DNA was used per 10 cm dish, and 15 µg of DNA was used per 15 cm dish. Proteins were given 2 to 3 days of expression before the samples were used.

## Plasmids

pMT-HA-Ub was a generous gift from Jianhang Jia (*Zhang et al., 2006*).

Scar (Berkley Expression Clone Collection FMO14142) (*Yu et al., 2011*).

Scar-Flag was created using quick change mutagenesis of FMO14142 following the manufacturer's protocol.

Prmha3-Non-stop-2XFlag-2XHA (ubiquitin-specific protease nonstop [*Drosophila melanogaster*] Sequence ID: AAD53181.1) (*Poeck et al., 2001*).

Prmha3-Non-stop-2XFlag-2XHA-0FA, Prmha3-Non-stop-2XFlag-2XHA-1FA, Prmha3-Non-stop-2XFlag-2XHA-2FA, Prmha3-Non-stop-2XFlag-2XHA-3FA, Prmha3-Non-stop-2XFlag-2XHA-WIRSFA, Prmha3-Non-stop-2XFlag-2XHA-C406A, Prmha3-Non-stop-2XFlag-2XHA-C406S were created using

quick change mutagenesis using Prmha3-Non-stop-2XFlag-2XHA as a template following the manufacturer's directions.

Primer list for quickchange mutagenesis:

SCAR-Flag F: GATGACGACAAGGTCAAACTTGCTGCTTAGACTAGTTCTAGT
SCAR-Flag R: ACTAGAACTAGTCTAAGCAGCAAGTTTGACCTTGTCGTCATC
WIRS0 phenylalanine to alanine F: GCAGTGGCCGAAGCGCCGGCAGGGGAACGGAACGGTGGGC
WIRS0 phenylalanine to alanine R: CCGTTCCCCTGCCGGCGCTTCGGCCACTGCTGCTGCTGC
WIRS1 phenylalanine to alanine F: CAGCTACGATACAGCCCGGGTCATCGACGCCTACTTCGCTGCTTGCG
WIRS1 phenylalanine to alanine R: GGCGTCGATGACCCGGGCTGTATCGTAGCTGTGCTCCTTCACATAGC
WIRS2 phenylalanine to alanine F: CCAGCGTGGTGTCGGCCCATTTGAAACGCTTCGAGCACTCAGCTCTG
WIRS2 phenylalanine to alanine R: CGAAGCGTTTCAAATGGGCCGACACCACGCTGGGCAGAGTGCGCAG
WIRS3 phenylalanine to alanine F: CGCAAGATCTCCTCGGCCATTCAATTCCCCGTGGAGTTCGACATG
WIRS3 phenylalanine to alanine R: CCACGGGGAATTGAATGGCCGAGGAGATCTTGCGATCGATCAGAGC
Non-stop C406A F: CTTAATCTGGGCGCCACTGCCTTCATGAACTGCATCGTC
Non-stop C406A R: GACGATGCAGTTCATGAAGGCAGTGGCGCCCAGATTAAG
Non-stop C406S F: CTTAATCTGGGCGCCACTAGCTTCATGAACTGCATCGTC
Non-stop C406S R: GACGATGCAGTTCATGAAGCTAGTGGCGCCCAGATTAAG

## RNAi treatment of BG3 cells

Short dsRNA was produced by amplifying cDNA with primers that contained T7 promoter sequences on the ends. These PCR products were then used as a template for *in vitro* transcription using the Ampliscribe T7 high yield transcription kit (epicentre catalog number: AS3107) following the manufacturer's instructions. Transcription was allowed to continue for 4 hr. To make the RNA double stranded it was then heated to 65°C for 30 min and cooled to room temperature. The RNA was then ethanol precipitated in order to clean it.

BG3 cells were plated at $2 \times 10^6$ cells/ml in six well dishes. Cells were allowed to adhere (approximately 4 hr) then treated with 10 µg of dsRNA. The cells were harvested for western blot after being soaked for 6 days and after 5 days for RNA analysis.

For 10 cm dishes cells were plated at $16 \times 10^6$ cells in 4 ml of serum-free media 120 µg of dsRNA was immediately added. Cells were incubated for 1 hr at 25°C. After 1 hr 9 ml of complete media was added. The cells were harvested for immunoprecipitation after being soaked for 6 days in dsRNA.

## RNAi

Atxn7 DRSC23138, Atxn7 DRSC35628, non-stop BKN21994, non-stop DRSC11378.
dsRNA-LacZ-R:          GCTAATACGACTCACTATAGGCCAAACATGACCARGATTACGCCAAGCT
dsRNA-LacZ-F: GCTAATACGACTCACTATAGGCCAAACGTCCCATTCGCCATTCAGGC

## Immunofluorescence

Cells were plated directly onto coverslips in six-well dishes and allowed to adhere. BG3 cells were fixed in 4% methanol-free formaldehyde after 2 days of growth. Cells were washed in 1xPBS and then either stored in 1xPBS for up to 1 week or immediately stained. Coverslips were washed with 1xPBS containing 0.5% Triton X-100 four times for 10 min. Coverslips were blocked in 5% BSA and 0.2% TWEEN 20 diluted in 1xPBS for 1 hr at room temperature. They were then incubated in primary antibody diluted in blocking buffer overnight at 4°C. The following day they were washed with 1xPBS containing 0.5% Triton X-100 four times for 10 min. They were then incubated in secondary antibody diluted 1:1000 in blocking buffer for 4hr at room temperature. Cells were then washed four times 10 min at room temperature in 1xPBS containing 5% Triton X-100. Coverslips were dried and mounted in Vectashield containing DAPI (Vector Labs catalog number: H-1200).

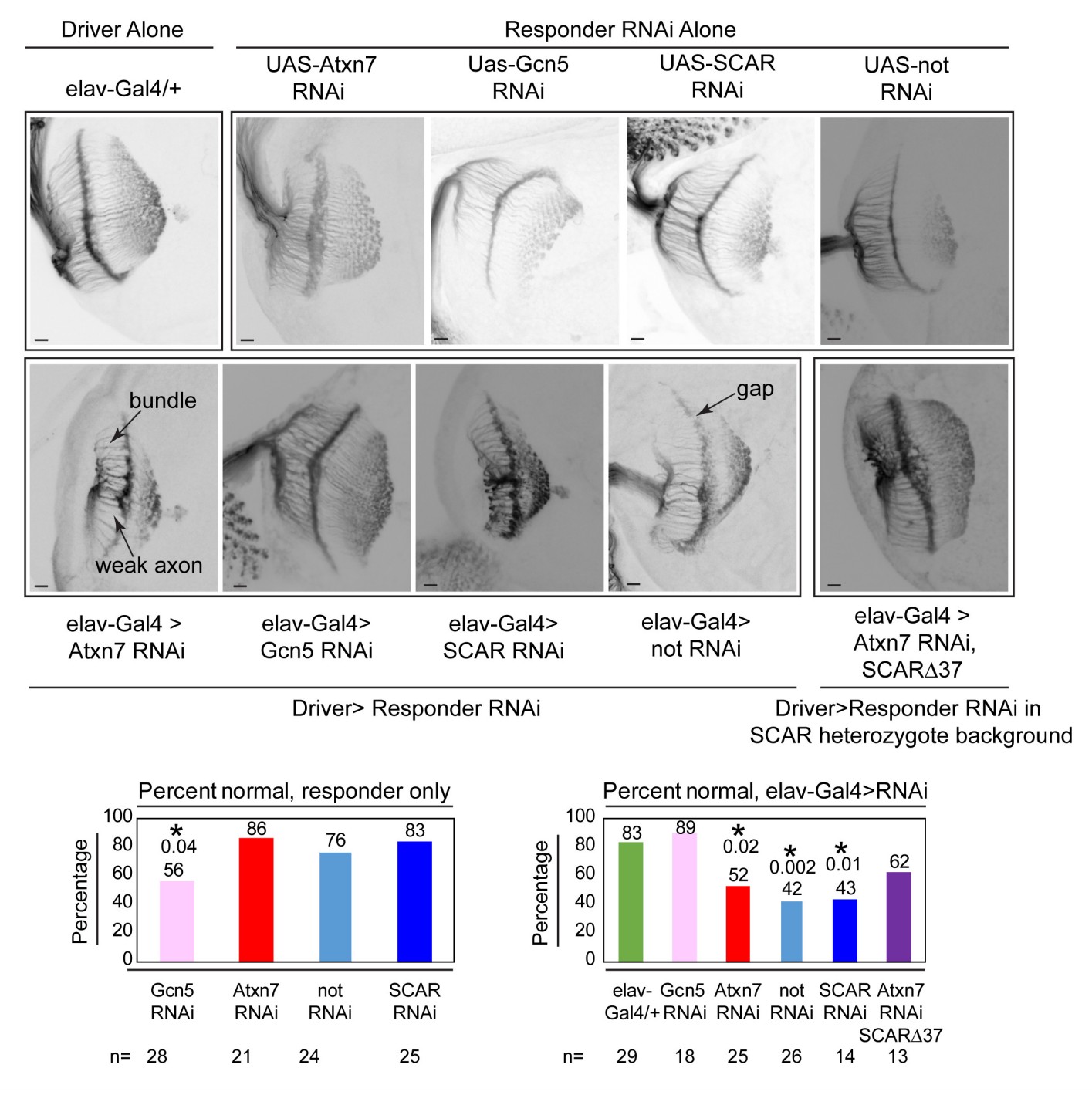

**Figure 8.** Chaoptin staining in the optic lobe of third instar larval brains reveals similar defects in *Atxn7*, *SCAR*, and *not* knockdowns *in vivo*. The elav-Gal4 driver was used to drive UAS-RNAi lines in larvae. Third instar larval brains were dissected and immunostained with a Chaoptin antibody. Images were adjusted with contrast limited adaptive histogram equalization using the ImageJ CLAHE algorithm (*Zuiderveld, 1994*). Optic lobes were analyzed for the presence of bundles, gaps, or weak axons. Examples of these are indicated in the figure. A brain was determined to be normal if it had three or fewer of these defects. The percentage of normal brains are shown for each genotype. A fisher's exact test was used to compare samples to the driver alone (elav-gal4/+). Significant p-values are marked with an asterisk. Error bars shows standard error.

DOI: https://doi.org/10.7554/eLife.49677.010

Larval brains were dissected into 4% methanol free formaldehyde/1xPBS and fixed at 4°C for 1 hr. Fixative was removed and the brains were washed with 1xPBS and then either stored at 4°C in

1XPBS for up to 1 week or immediately stained. Brains were washed three times in 1xPBS containing 5% Triton X-100. They were then incubated in 1xPBS containing 5% Triton for 20 min at room temperature. Cells were blocked for 1 hr at room temperature in blocking buffer containing 0.2% TWEEN 20% and 5% BSA diluted in 1xPBS. Brains were then incubated in Alexafluor conjugated phalloidin diluted 1:20 in blocking buffer overnight at 4°C. They were then wash four times for 10 min at room temperature in 1XPBS containing 5% Triton X-100. Larval brains were mounted in Vectashield containing DAPI (Vector Labs catalog number: H-1200). For Chaoptin staining, brains were incubated in primary antibody diluted in blocking buffer at 4°C overnight following blocking. They were washed four times 10 min at room temperature in 1xPBS containing 5% Triton X-100. They were then incubated in secondary antibody diluted 1:1000 in blocking buffer overnight at 4°C. They were then washed four times 10 min in 1xPBS containing 5% Triton. Larval brains were mounted in Vectashield containing DAPI (Vector Labs catalog number: H-1200).

A Zeiss LSM-5 Pascal upright microscope and LSM software was used for imaging. Cells were imaged with a 40X objective aperture number 1.30 oil or a 63X objective aperture number 1.25 Oil and Z-stacks were acquired using LSM software with a slice every 1 μm. Brains were imaged with a 20X objective aperture number 0.45 air and Z stacks were taken with slices every 4 μm. All images were taken at room temperature. All settings remained the same within an experiment.

## Antibodies

Mouse anti-SCAR (DSHB P1C1-SCAR RRID:AB_2618386) was used at a dilution of 1:100. Rat anti HA (Roche catalog number: 11867423001 RRID:AB_390918) was used at a dilution of 1:1000. Guinea pig anti-Non-stop (*Mohan et al., 2014a*) was used at a dilution of 1:150. Mouse anti-Chaoptin (deposited to the DSHB by Benzer, S and Colley, N.) (DSHB 24B10 RRID:AB_528161) was used at a dilution of 1:250. Goat anti-Rat IGG −568 (Invitrogen Catalog number: A-11077 RRID:AB_141874) was used at a dilution of 1:1000. Goat anti-Mouse IGG-488 (Invitrogen Catalog number: A-11001 RRID:AB_2534069) was used at a dilution of 1:1000. Goat anti-Guinea pig IGG- 488 (Invitrogen Catalog number: A11073 RRID:AB_142018) was used at a dilution of 1:1000. Phalloidin-488 (Invitrogen Catalog number: A12379 RRID:AB_2759222) was used at a dilution of 1:20. Phalloidin-568 (Invitrogen Catalog number: A12380 RRID:AB_2810839) was used at a dilution of 1:20.

## Image analysis

Stacks were exported as TIFFs from the LSM software and analyzed in FIJI (*Schindelin et al., 2012*) RRID:SCR_002285. For analysis z projections were created by sum or max.

To analyze cell shape and phalloidin structures in BG3 cells the phalloidin images were thresholded in ImageJ and used to create ROIs. The number of projections was counted by hand. A t-test was used to test the significance between the mock transfected control and the Non-stop-2xFlag-2 transfected cell lines in cellular shape and the number of projections.

To measure SCAR levels in BG3 cells projections were made by sum and ROIs were drawn by hand around the entirety of the scar staining and integrated density was measured for both HA and SCAR. For BG3 cells, a second ROI was drawn around the nuclei by thresholding the DAPI image and the SCAR intensity and HA intensity in the nucleus was measured. A t-test was used to compare between treated and untreated cells.

To measure phalloidin intensity in larval brains, max projections were created and ROIs were drawn by hand. Brains were only measured if they were not mounted at an angle that excluded part of the VNC and the VNC was intact. Thresholding and image calculator were used to subtract background. A t-test was used to compare differences between genotypes.

To measure integrity of the optic lobe max projections were created and images were thresholded identically using contrast limited adaptive histogram equalization using the ImageJ CLAHE algorithm (*Zuiderveld, 1994*) and the number of gaps and bundles were counted by hand. Fisher's exact test was used to compare differences between genotypes.

## *Ex vivo* culturing of third instar brains

*Ex vivo* culturing of larval brains was performed as described (*Rabinovich et al., 2015*; *Prithviraj et al., 2012*). Third instar brains were dissected in 1xPBS and quickly placed into culture media (Schneider's insect media supplemented with 10% FBS, 1% pen/strp, 1:10,000 insulin, and 2

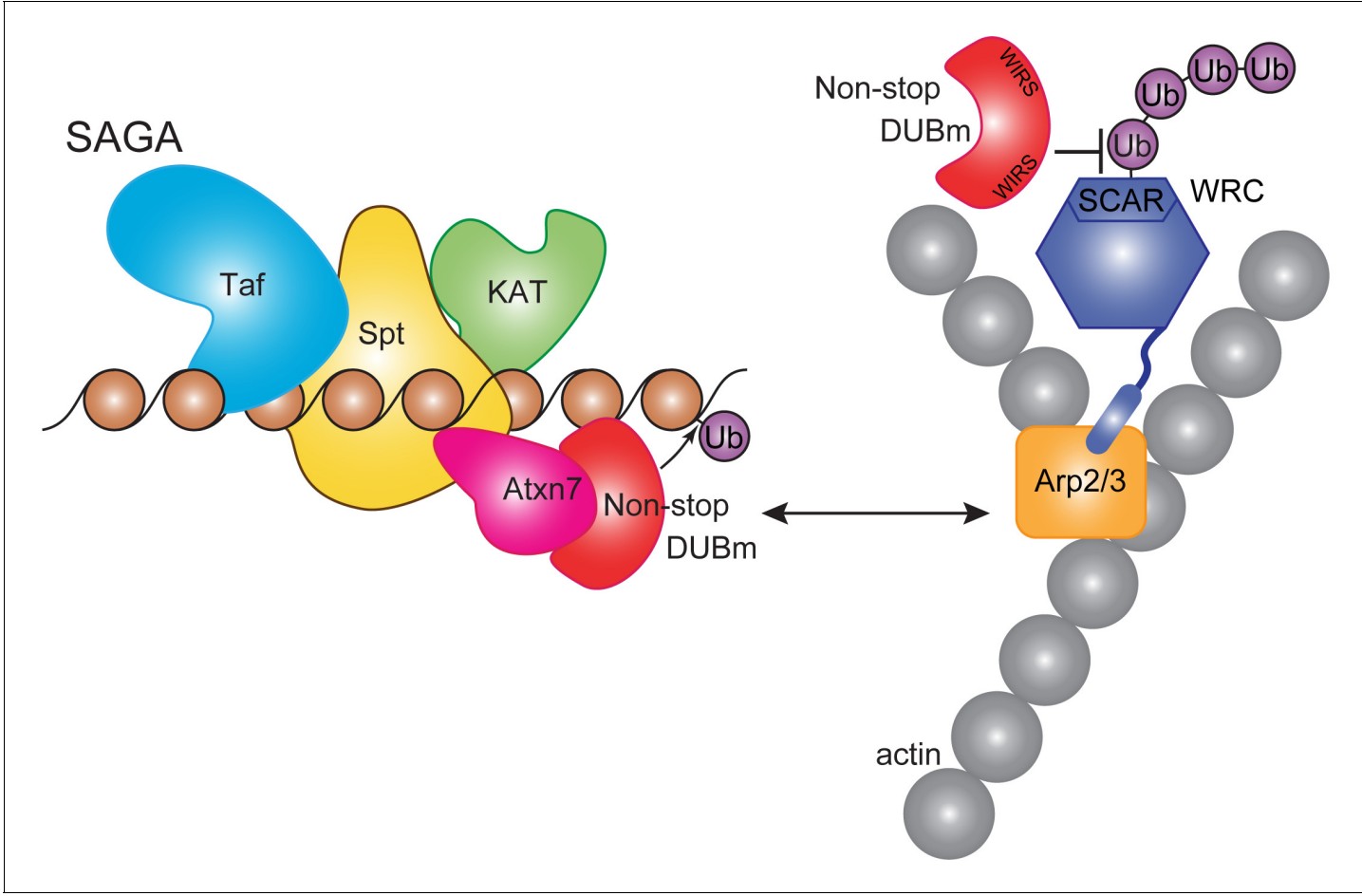

**Figure 9.** Non-stop regulates SCAR protein levels and location. Model showing Non-stop interacts with WRC in an Atxn7-dependent manner, where Non-stop then counteracts degradation of WRC subunit SCAR. Loss of Atxn7 leads to increased availability of Non-stop for interaction with WRC.
DOI: https://doi.org/10.7554/eLife.49677.011

µg/ml ecdysone) in 24-well dish. The dish was wrapped in parafilm and incubated at 25°C. After 6 hr of culturing, the media was changed. The wells receiving MG132 then had 50 µM of MG132 added. Brains were collected for protein samples after 24 hr.

## Protein sample preparation

All western blots using tissue culture cells were done from cells collected from a six-well dish. The media was removed and 200 µl of 2xUREA sample buffer (50 mM Tris HCl pH 6.8, 1.6% SDS, 7% glycerol, 8M Urea, 4% β mercaptoethanol, 0.016% bromophenol blue) and 2 µl of 10 µg/ml benzonase was added to the dish. After the mixture was lysed they were boiled at 80°C for 10 min.

Larval brains were dissected into 1xPBS. The PBS was removed and 2xUREA sample buffer was added in a 1:1 ratio and 10 units of Benzonase was added. The brains were homogenized by pipetting and boiled at 80°C for 10 min.

## Immunoprecipitation

### Non-denaturing extract

Transfected cells were resuspended in Extraction Buffer (20 mM HEPES (pH7.5), 25% Glycerol, 420 mM NaCl, 1.5 mM MgCl$_2$, 0.2 mM EDTA, 1:100 ethidium bromide with protease inhibitors added. 1% NP-40 was added and the cells were pipetted up and down until the solution was homogenous. They were placed on ice for 1 hr with agitation every 10–15 min. They were then centrifuged for 30 min at 4°C at 20,000 x g.

The supernatant was placed into a new tube and stored at – 80℃ for up to 1 week. Lysates were thawed and 36 µg of protein was removed for input and mixed with 2 x Urea sample buffer (50 mM Tris-HCl (pH 6.8), 1% SDS, 7% Glycerol, 8 M Urea, 4% beta-mercapto-ethanol, 0.016% Bromophenol blue) and boiled for ten minutes at 80℃. An equal volume of Dignum A buffer (10 mM HEPES (pH 7.5), 1.5 mM MgCl$_2$, 10 mM KCl) was added to the lysates in order to adjust the salt concentration to 210 mM NaCl. The entire volume was added to 15 µl of Mouse IGG agarose (Sigma Catalog number: A0919) that had been equilibrated in one volume extraction buffer plus one volume dignum A. They were then rotated at 4℃ for 1 hr. After 1 hr, they were centrifuged for 1 min at 4℃ at 4500 x g. The supernatant was then added to anti-FLAG M2 Affinity Gel (Sigma- Aldrich Catalog number: A2220, RRID:AB_10063035) that had been equilibrated in one volume extraction buffer plus one volume Dignum. They were then rotated for 1 hr at 4℃. After 1 hr, they were centrifuged at 4500 x g at 4℃ for 30 s. The supernatant was removed and the gel was washed five times in extraction buffer plus dignum A. The resin was then resuspended in 40 µl 2XUREA sample buffer and boiled for 10 min at 80℃.

### In vitro ubiquitination assays

Immunoprecipitations with the intention to immunoblot for ubiquitin were prepared in denaturing extracts as described (*Zhang et al., 2006*; *Wang and Li, 2006*). The day of harvesting cells were treated with 50 µM of MG132 for 6 hr. After 6 hr, cells were scraped and resuspended in Denaturing Buffer (1% SDS/50 mM Tris [pH 7.5], 0.5 mM EDTA/1 mM DTT) plus the protease inhibitors PMSF, sodium orthovanadate, Aprotinin, Leupeptin, Pepstatin A, Sodium butyrate, and Sodium fluoride. They were boiled at 100℃ for 5 min. They were then diluted 10 fold in dilution buffer (10 mM Tris-HCl, pH 8.0, 150 mM NaCl, 2 mM EDTA, 1% Triton), mixed well, and then centrifuged for 30 min at 4℃ at 20,000 x g. The supernatant was then added to 15 µl anti-FLAG M2 Affinity Gel (Sigma-Aldrich Catalog number: A2220, RRID:AB_10063035) that had been equilibrated dilution buffer. They were then rotated for 1 hr at 4℃. After 1 hr, they were centrifuged at 4000 x g at 4℃ for 30 s. The supernatant was removed and the gel was washed four times with washing buffer (10 mM Tris-HCl, pH 8.0, 1 M NaCl, 1 mM EDTA, 1% NP-40). The resin was then resuspended in 33 µl 2XUREA sample buffer and boiled for 10 min at 80℃.

## Endogenous Non-stop immunopreciptiation

Larva were placed into Extraction Buffer (20 mM HEPES (pH7.5), 25% Glycerol, 420 mM NaCl, 1.5 mM MgCl$_2$, 0.2 mM EDTA, and 1:100 ethidium bromide with protease inhibitors added. 1% NP-40 was added and the cells were pipetted up and down until the solution was homogenous. Samples were spun for 30 min at 20,000 xg at 4℃. 36 µg of protein was removed for input and mixed with 2 x Urea sample buffer (50 mM Tris-HCl (pH 6.8), 1% SDS, 7% Glycerol, 8 M Urea, 4% beta-mercapto-ethanol, 0.016% Bromophenol blue), then boiled for 10 min at 80℃. Input was run on a gel and a western blot for Non-stop was performed to determine relative Non-stop concentrations and used to normalize the amount used in pull down. An equal volume of dignum A buffer (10 mM HEPES (pH 7.5), 1.5 mM MgCl$_2$, 10 mM KCl) was added to the lysates in order to adjust the salt concentration to 210 mM NaCl. Extracts were centrifuged for 30 min at 4℃ at 20,000 x g. The Non-stop antibody was added at a concentration of 1:50 and extracts were rotated at 4℃ overnight. Extracts were added to 50 µl of protein A dynabeads (Life technologies: 10002D) equilibrated in Extraction Buffer and rotated at 4℃ for 2 hr. The extract was removed and the dynabeads were washed four times in Extraction Buffer with 1% NP-40. Bead were resuspended in 30 µl of 2XUREA sample buffer as above and boiled for 10 min at 80℃.

## Western blot

Protein samples were run on either 6% polyacrylamide gel (Immunoprecipitation) or on 8% polyacrylamide gels (all other samples) in 1X Laemli electrode buffer. Proteins were transferred to Amersham Hybond P 0.45 µM PVDF membrane (catalog number: 10600023) using the Trans-blot turbo transfer system from BioRad. The semi-dry transfer took place in Bjerrum Schafer-Nielsen Buffer with SDS (48 mM Tris, 39 mM Glycine, 20% Methanol, 0.1% SDS) at 1 Amp 25 V for 20 min. Membranes were stained in Ponceau (0.2% Ponceau S, 2% Acetic Acid) for 20 min at room temperature as a loading control. The membrane was then briefly washed in 1XPBS and imaged on a Chemidoc MP Imaging

system. Membranes were then washed three times for 5 min to remove excess ponceau. Membranes were then blocked for 1 hr in 5% non-fat dried milk diluted in 0.05% TWEEN-20 in 1XPBS. The membrane was then briefly rinsed and incubated in primary antibody diluted in 1% non-fat dried milk diluted in 0.05% TWEEN-20 for either 1 hr at room temperature or overnight at 4°C. The membrane was then washed four times for 5 min in 1% TWEEN-20 diluted in 1XPBS. The membrane was then incubated for 1 hr at room temperature in secondary antibody diluted 1:5000 in 1% non-fat dried milk diluted in 0.05% Triton-X 100. The membrane was then washed four times for 5 min in 1% TWEEN-20 diluted in 1xPBS. Bioworld ICL (catalog number: 20810000–1) was used to visualize the proteins following the manufacturer's instruction. The chemiluminescence was imaged on a Chemidoc MP Imaging system. Band intensities were measured on ImageLab software.

## Antibodies

Mouse anti-SCAR (DSHB P1C1-SCAR, RRID: AB_2618386) was used at a dilution 1:250. Rat anti HA-HRP (Roche catalog number: 12013819001, RRID:AB_390917) was used at a dilution of 1:500. Guinea Pig anti-Non-stop (*Mohan et al., 2014a*) was used at a dilution of 1:1000. Guinea Pig anti-Ada2b (*Kusch et al., 2003*) was used at a dilution of 1:1000. Rabbit anti-Atxn7 (*Mohan et al., 2014a*) was used at a concentration of 1:2000. Goat anti Guinea Pig HRP (Jackson ImmunoResearch INC catalog number: 106-035-003, RRID:AB_2337402) was used at a dilution of 1:10,000. Goat anti-mouse HRP (Jackson ImmunoResearch INC catalog number: 115-035-003, RRID:AB_10015289) was used at a dilution of 1:5000. Goat anti-Rabbit HRP (Jackson ImmunoResearch INC catalog number: 111-035-003, RRID:AB_2313567) was used at a dilution of 1:10,000.

## QPCR

RNA was extracted using the Invitrogen PureLink RNA mini kit (Fisher catalog number: 12 183 018A) following the manufacturer's instructions. The optional on column DNA digest was performed using the PureLink DNase set (Fisher Catalog number: 12185010). The high-capacity cDNA reverse transcription kit (ThermoFisher catalog number: 4374966) was used to create cDNA using the manufacturer's protocol. 3 µl of cDNA created from 1 µg of RNA was used in downstream qPCR analysis. TaqMan gene expression assays were run following the manufacturer's directions. PCR was performed using the TaqMan Universal PCR Master Mix (ThermoFisher catalog number: 4364340). The reactions were run on an ABI 7500 Real-time PCR machine.

## TaqMan assay IDs

RpL32 Dm02151827_g1, SCAR Dm01810606_g1, Not Dm01823071_g1, Atxn7 Dm01800874_g1.

## Fly strains

SCAR [Delta37]: w[*]; SCAR[Delta37] P(ry[+t7.2]=neoFRT)40A/CyO, P(w[+mC]=GAL4 twi.G)2.2, P(UAS-2xEGFP)AH2.2. (BDSC catalog number: 8754 RRID:BDSC_8754).

Atxn7: y[1] w[67c23]; P(y[+mDint2] w[BR.E.BR]=SUPor P)CG9866[KG02020]/CyO, P(w[+mC]=GAL4 twi.G)2.2, P(UAS-2xEGFP)AH2.2. (BDSC catalog number: 14255, RRID:BDSC_14255).

Non-stop: P(ry[+t7.2]=PZ)not[02069] ry[506]/TM3, P(w[+mC]=GAL4 twi.G)2.3, P(UAS-2xEGFP)AH2.3, Sb[1] Ser[1]. (BDSC catalog number: 11553 RRID:BDSC_11553).

Wild type: Oregon-R (DGGR Catalog number: 109162, RRID:DGGR_109612).

Elav-Gal4: P(w[+mW.hs]=GawB)elav[C155] (BDSC catalog number: 458, RRID:BDSC_458).

Uas-Atxn7 RNAi: P{KK110634} VIE-260B (VDRC stock number: 102078, RRID:FlyBase_FBst0473949).

Uas-Gcn5 RNAi: w[1118]; P{GD11218}v21786 (VDRC stock number: 21786. RRID:FlyBase_FBst0454233).

Uas-SCAR RNAi: y[1] sc[*] v[1]; P(y[+t7.7] v[+t1.8]=TRiP.HMS01536)attP40 (BDSC catalog number: BL36121, RRID:BDSC_36121).

Uas-Not RNAi: y[1] v[1]; P(y[+t7.7] v[+t1.8]=TRiP.JF03152)attP2/TM6B, Tb (BDSC catalog number: 28725 rebalanced to Tm6b, Tb, RRID:BDSC_28725).

Uas-Atxn7 RNAi, SCARΔ37: w[*]; SCAR[Delta37] P(ry[+t7.2]=neoFRT)40A/CyO, P(w[+mC]=GAL4 twi.G)2.2, P(UAS-2xEGFP)AH2.2.; P{KK110634}y VIE-260B/TM6C, cu[1] Sb[1] Tb[1].

## Multidimensional protein identification technology

TCA-precipitated protein pellets were solubilized using Tris-HCl pH 8.5 and 8 M urea, followed by addition of TCEP (Tris(2-carboxyethyl)phosphine hydrochloride; Pierce) and CAM (chloroacetamide; Sigma) to a final concentration of 5 mM and 10 mM, respectively. Proteins were digested using Endoproteinase Lys-C at 1:100 w/w (Roche) at 37°C overnight. The samples were brought to a final concentration of 2 M urea and 2 mM $CaCl_2$ and a second digestion was performed overnight at 37°C using trypsin (Roche) at 1:100 w/w. The reactions were stopped using formic acid (5% final). The digested size exclusion eluates were loaded on a split-triple-phase fused-silica micro-capillary column and placed in-line with a linear ion trap mass spectrometer (LTQ, Thermo Scientific), coupled with a Quaternary Agilent 1100 Series HPLC system. The digested Not and control FLAG-IP eluates were analyzed on an LTQ-Orbitrap (Thermo) coupled to an Eksigent NanoLC-2D. In both cases, a fully automated 10-step chromatography run was carried out. Each full MS scan (400–1600 m/z) was followed by five data-dependent MS/MS scans. The number of the micro scans was set to one both for MS and MS/MS. The settings were as follows: repeat count 2; repeat duration 30 s; exclusion list size 500 and exclusion duration 120 s, while the minimum signal threshold was set to 100.

## Mass spectrometry data processing

The MS/MS data set was searched using ProLuCID (v. 1.3.3) against a database consisting of the long (703 amino acids) isoform of non-stop, 22,006 non-redundant *Drosphila melanogaster* proteins (merged and deduplicated entries from GenBank release 6, FlyBase release 6.2.2 and NCI RefSeq release 88), 225 usual contaminants, and, to estimate false discovery rates (FDRs), 22,007 randomized amino acid sequences derived from each NR protein entry. To account for alkylation by CAM, 57 Da were added statically to the cysteine residues. To account for the oxidation of methionine to methionine sulfoxide, 16 Da were added as a differential modification to the methionine residue. Peptide/spectrum matches were sorted and selected to an FDR less than 5% at the peptide and protein levels, using DTASelect in combination with swallow, an in-house software. The permanent URL to the dataset is: ftp://massive.ucsd.edu/MSV000082625. The data is also accessible from: ProteomeXChange accession: PXD010462 http://proteomecentral.proteomexchange.org/cgi/GetDataset?ID=PXD010462. MassIVE | Accession ID: MSV000082625 - ProteomeXchange | Accession ID: PXD010462. Original mass spectrometry data underlying this manuscript can also be accessed from the Stowers Original Data Repository at http://www.stowers.org/research/publications/libpb-1242.

## Acknowledgements

Jerry L Workman and Susan M Abmayr for advice and guidance. Jakob Waterborg for contributing plasticware reagents and small equipment. Boyko Atanassov and Stone (Bayou) Chen for critical reading of early draft manuscripts. University of Missouri Faculty Scholars training program. Funding agencies – School of Biological Science, UMKC; University of Missouri Research Board; UMKC Students Engaged in the Arts and Research (SEARCH); UMKC Summer Undergraduate Research Opportunity (SUROP) scholars programs; funds provided by the University of Missouri – Kansas City Provost Strategic Initiatives Fund through Project ADVANCER - Academic Development Via Applied and Cutting-Edge Research for UMKC undergraduate and professional students from underrepresented minorities Program; and NIGMS grant 5R35GM118068 to JL Workman. The Drosophila Genomics Resource Center (NIH Grant 2P40OD010949) for reagents. The mouse anti SCAR (deposited by Susan Parkhurst and mouse anti Chaoptin antibody (deposited by Benzer, S and Colley, N) were obtained from the Developmental Studies Hybridoma Bank, created by the NICHD of the NIH and maintained at The University of Iowa, Department of Biology, Iowa City, IA 52242. Transgenic fly stocks and/or plasmids were obtained from the Vienna Drosophila Resource Center (VDRC, www.vdrc.at). Stocks obtained from the Bloomington Drosophila Stock Center (NIH P40OD018537) were used in this study.

# Additional information

## Funding

| Funder | Grant reference number | Author |
| --- | --- | --- |
| National Institute of General Medical Sciences | 5R35GM118068 | Ryan D Mohan |
| University of Missouri-Kansas City | Provost Strategic Initiatives Fund through Project ADVANCER - Academic Development Via Applied and Cutting-Edge Research for UMKC undergraduate and professional students from underrepresented minorities Program | Sarah R Rapp |
| University of Missouri-Kansas City | Provost Strategic Initiatives Fund through Project ADVANCER - Academic Development Via Applied and Cutting-Edge Research for UMKC undergraduate and professional students from underrepresented minorities Program | Jarrid L Jack |
| University of Missouri-Kansas City | Provost Strategic Initiatives Fund through Project ADVANCER - Academic Development Via Applied and Cutting-Edge Research for UMKC undergraduate and professional students from underrepresented minorities Program | Pedro Morales-Sosa |
| University of Missouri-Kansas City | UMKC Students Engaged in the Arts and Research (SEARCH) | Ada Thapa |
| University of Missouri-Kansas City | UMKC Students Engaged in the Arts and Research (SEARCH) | Sara A Miller |
| University of Missouri-Kansas City | UMKC Students Engaged in the Arts and Research (SEARCH) | Paige M Gerhart |
| University of Missouri-Kansas City | UMKC Summer Undergraduate Research Opportunity (SUROP) scholars programs | Ada Thapa |
| University of Missouri-Kansas City | UMKC Summer Undergraduate Research Opportunity (SUROP) scholars programs | Pedro Morales-Sosa |
| University of Missouri-Kansas City | UMKC Funding for Faculty Excellence | Ryan D Mohan |
| University of Missouri Research Board | | Ryan D Mohan |

The funders had no role in study design, data collection and interpretation, or the decision to submit the work for publication.

## Author contributions

Veronica Cloud, Conceptualization, Data curation, Supervision, Validation, Investigation, Visualization, Methodology, Project administration, Writing—review and editing; Ada Thapa, Pedro Morales-Sosa, Tayla M Miller, Daniel Holsapple, Paige M Gerhart, Elaheh Momtahan, Jarrid L Jack,

Edgardo Leiva, Sarah R Rapp, Lauren G Shelton, Richard A Pierce, Skylar Martin-Brown, Investigation, Assisted with preparing the manuscript; Sara A Miller, Supervision, Investigation; Laurence Florens, Conceptualization, Resources, Data curation, Software, Formal analysis, Supervision, Validation, Investigation, Visualization, Methodology, Project administration, Writing— review and editing; Michael P Washburn, Resources, Data curation, Software, Formal analysis, Supervision, Validation, Investigation, Visualization, Methodology, Project administration, Writing— review and editing; Ryan D Mohan, Conceptualization, Resources, Data curation, Formal analysis, Supervision, Funding acquisition, Validation, Investigation, Visualization, Methodology, Writing— original draft, Project administration, Writing—review and editing

#### Author ORCIDs
Veronica Cloud https://orcid.org/0000-0003-3701-4150
Michael P Washburn http://orcid.org/0000-0001-7568-2585
Ryan D Mohan https://orcid.org/0000-0002-7624-4605

#### Decision letter and Author response
Decision letter https://doi.org/10.7554/eLife.49677.016
Author response https://doi.org/10.7554/eLife.49677.017

## Additional files

#### Supplementary files
• Transparent reporting form
DOI: https://doi.org/10.7554/eLife.49677.012

#### Data availability
Original mass spectrometry data underlying this manuscript can be accessed from ProteomeX-Change at http://proteomecentral.proteomexchange.org/cgi/GetDataset?ID=PXD010462 and from the Stowers Original Data Repository at http://www.stowers.org/research/publications/libpb-1242.

The following dataset was generated:

| Author(s) | Year | Dataset title | Dataset URL | Database and Identifier |
|---|---|---|---|---|
| Skylar Martin-Brown, Laurence Florens, Michael P Washburn, Ryan D Mohan | 2019 | Data from: Ataxin-7 and Non-stop coordinate SCAR protein levels, subcellular localization, and actin cytoskeleton organization | http://proteomecentral. proteomexchange.org/ cgi/GetDataset?ID= PXD010462 | ProteomeXchange, PXD010462 |

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
