## [Decision Letter]

Thank you for submitting your article "Ataxin-7 and Non-stop coordinate SCAR protein levels, subcellular localization, and actin cytoskeleton organization" for consideration by *eLife*. Your article has been reviewed by three peer reviewers, including Irwin Davidson as the Reviewing Editor and Reviewer #1, and the evaluation has been overseen by Kevin Struhl as the Senior Editor. The following individual involved in review of your submission has agreed to reveal their identity: François Payre (Reviewer #2).

The reviewers have discussed the reviews with one another and the Reviewing Editor has drafted this decision to help you prepare a revised submission.

Summary:

This manuscript investigates the potential non-histone substrates of non-stop and the deubiquitinase module of the *Drosophila* SAGA complex. Cell lines expressing tagged derivatives of Atxn7 and Non-stop were generated and purified by tandem affinity purification followed by size exclusion chromatography and partner proteins in enzymatically active fractions were identified by mass-spectrometry. Using this approach, the authors identify Scar as an interaction partner and potential substrate for non-stop Dub activity and they identify a conserved WIRS sequence motif that mediates interactions between Non-stop and Scar.

While the reviewers found the results of this study novel and interesting several major issues have to be addressed in a revised version.

Essential revisions:

1) The authors show that Non-stop interacts with Scar and that modulating Non-stop levels influences those of Scar. Nevertheless, this falls short of a demonstration that Non-stop acts as a Dub for Scar. The authors therefore should perform experiments with tagged ubiquitin to follow ubiquitination of SCAR and H2B in conditions of ataxin-7 and Non-stop manipulation with or without proteasome inhibition. In the same vein, testing a catalytic-dead version of Non-stop and versions of Non-stop where the WIRS motifs are mutated would also be pertinent in these assays.

2) It is essential that the authors address the effects of mutating the WIRS motifs on interaction with Scar directly by co-immunoprecipitation and immunoblot. They should also investigate uncoupling of Non-stop functions in SAGA vs WAVE complexes. Testing the influence of individual (or combined) mutation of WIRS in the biochemical interaction with Ataxin-7 and SCAR could address this. Also, the authors show by western blot that knockdown of Atxn7 increases SCAR protein levels. Does this increased SCAR protein form complexes with endogenous Non-stop that are detectable by immunoprecipitation? According to the model that the authors propose, SCAR and Non-stop would be expected to co-precipitate at a higher level in the Atxn7 knockdown background compared to wildtype.

3) The authors should show immunoblots to assess the levels of overexpressed tagged proteins in the S2 cells versus the endogenous proteins to address whether strong overexpression could induce artificial interactions. After tandem IP, the authors should perform SDS PAGE and Coomassie blue/silver nitrate staining to assess the composition of the precipitated complexes and if possible they should perform SDS PAGE after size exclusion chromatography to assess the complexity in the composition of each fraction. At present, the reader cannot assess the purity of the samples, or the stoichiometry of the different complexes. It would also be useful to track the elution of the tagged bait at each stage and in the fractions from the size exclusion column. Which proportion of the bait Non-stop or Atxn7 proteins are in the different fractions?

4) The authors should provide higher quality images to demonstrate co-localization of SCAR and Non-Stop.

5) The functional consequences of discovering this novel interactor of Non-stop remain barely addressed, being limited to gross examination of cultured cell morphology and F-actin distribution. In addition to Non-stop, null alleles of SCAR ataxin-7, sgf11 and gcn5 are available. The authors should assay in vivo whether some of the known Non-stop phenotypes (e.g. retinal degeneration, reduced locomotion, shortened life-time, or altered axon targeting of photoreceptors and migration of lamina glial cells) could be linked to altered SCAR levels, and suppressed upon reducing SCAR activity. Specifically, can they investigate the influence of the absence of Non-stop (vs other SAGA components), the effects of Non-stop overexpression and Atxatin-7 inactivation on SCAR levels and neural cell cytoskeletal organization. Can they investigate genetic interactions between non-stop and SCAR, for example suppression of some nonstop phenotypes upon reduction of SCAR function?

---

## [Author Response]

Essential revisions:1) The authors show that Non-stop interacts with Scar and that modulating Non-stop levels influences those of Scar. Nevertheless, this falls short of a demonstration that Non-stop acts as a Dub for Scar. The authors therefore should perform experiments with tagged ubiquitin to follow ubiquitination of SCAR and H2B in conditions of ataxin-7 and Non-stop manipulation with or without proteasome inhibition. In the same vein, testing a catalytic-dead version of Non-stop and versions of Non-stop where the WIRS motifs are mutated would also be pertinent in these assays.

We used the reviewer suggestions in addition to other new experiments to interrogate whether Non-stop was truly a SCAR DUB. Together, these new results further support the interpretation that Non-stop is indeed a SCAR DUB.

In Figure 4, we use binding site mutants of Non-stop which reduce, or increase Non-stop catalytic site affinity for ubiquitin to show that these bind to SCAR more and less, respectively (Figures 4A, 4B). We show that knockdown of Non-stop leads to increased ubiquitination of SCAR in BG3 cells (Figure 4C). Lastly, we show that Non-stop mutant larval brains have less SCAR, but this can be rescued by treating those brains with proteasome inhibitor MG132 (Figure 4D). Together, these results support a model where Non-stop counters ubiquitination and proteolytic degradation of SCAR.

2) It is essential that the authors address the effects of mutating the WIRS motifs on interaction with Scar directly by co-immunoprecipitation and immunoblot. They should also investigate uncoupling of Non-stop functions in SAGA vs WAVE complexes. Testing the influence of individual (or combined) mutation of WIRS in the biochemical interaction with Ataxin-7 and SCAR could address this. Also, the authors show by western blot that knockdown of Atxn7 increases SCAR protein levels. Does this increased SCAR protein form complexes with endogenous Non-stop that are detectable by immunoprecipitation? According to the model that the authors propose, SCAR and Non-stop would be expected to co-precipitate at a higher level in the Atxn7 knockdown background compared to wildtype.

We performed each of these experiments and the results support the proposed model.

– We mutated the WIRS motifs and show they reduce interaction with SCAR in pull-down experiments (Figure 6B, left).

– We investigated whether WIRS mutation could uncouple SCAR and SAGA functions and they did. While WIRS mutant Non-stop bound less to SCAR, it maintained interaction with Atxn7 (Figure 6B, right)

– We tested whether loss of Atxn7 resulted in detectable increased interaction between Non-stop and SCAR. Indeed it did. The interaction increased 1.4-fold (Figure 3A).

3) The authors should show immunoblots to assess the levels of overexpressed tagged proteins in the S2 cells versus the endogenous proteins to address whether strong overexpression could induce artificial interactions. After tandem IP, the authors should perform SDS PAGE and Coomassie blue/silver nitrate staining to assess the composition of the precipitated complexes and if possible they should perform SDS PAGE after size exclusion chromatography to assess the complexity in the composition of each fraction. At present, the reader cannot assess the purity of the samples, or the stoichiometry of the different complexes. It would also be useful to track the elution of the tagged bait at each stage and in the fractions from the size exclusion column. Which proportion of the bait Non-stop or Atxn7 proteins are in the different fractions?

These controls are now shown.

– The stable cell line expresses comparable levels of Non-stop to wild-type cell lines (Figure 1C).

– The silver stain analysis of the fractionated complexes, sent for mass spec, are shown (Figure 1D).

– The stoichiometry of the complexes is shown in the dNSAF value. It is a proxy for the amount of each subunit present in the purification (Figure 2A).

– The amount of Atxn7 and Non-stop in the groups of complexes is shown in the western blots of the eluted fractions (Figure 1E, upper).

The composition of the peak of Group 2 is shown (Figure 1E, right).

4) The authors should provide higher quality images to demonstrate co-localization of SCAR and Non-Stop.

These are now provided in all figures.

5) The functional consequences of discovering this novel interactor of Non-stop remain barely addressed, being limited to gross examination of cultured cell morphology and F-actin distribution. In addition to Non-stop, null alleles of SCAR ataxin-7, sgf11 and gcn5 are available. The authors should assay in vivo whether some of the known Non-stop phenotypes (e.g. retinal degeneration, reduced locomotion, shortened life-time, or altered axon targeting of photoreceptors and migration of lamina glial cells) could be linked to altered SCAR levels, and suppressed upon reducing SCAR activity. Specifically, can they investigate the influence of the absence of Non-stop (vs other SAGA components), the effects of Non-stop overexpression and Atxatin-7 inactivation on SCAR levels and neural cell cytoskeletal organization. Can they investigate genetic interactions between non-stop and SCAR, for example suppression of some nonstop phenotypes upon reduction of SCAR function?

We considered the intersection between SCAR and SAGA DUBm function as described in their respective literature. Misregulation of retinal axon targeting was common. To this end, we reduced SCAR gene copy number and knocked down Atxn7 to determine whether reducing SCAR could rescue disorganization of retinal axons. Although there was improvement when compared to brains which had two copies of SCAR, this failed to produce statistical significance, and so we conclude that SCAR reduction along cannot rescue axon mistargeting in vivo (Figure 8). This analysis also revealed that axon targeting is specially regulated by the SAGA DUBm.

However, we also performed the complimentary experiment and epistasis analysis. We examined whether loss of Atxn7 could rescue reductions in F actin that are observed in SCAR heterozygotes. In this case, heterozygosity of Atxn7 could rescue heterozygosity of SCAR (Figure 7C). Figures 7A and 7B support 7C, showing that loss of Non-stop decreases F actin and loss of Atxn7 increases it.